# KinemaDiff: Towards Diffusion for Coherent and Physically Plausible Human Motion Prediction

**Ye Lu**[1]*, **Jie Wang**[2]*, **Tianyi Liu**[1], **Jianjun Gao**[1], **Kim-Hui Yap**[1]
[1]Nanyang Technological University      [2]Beijing Institute of Technology

## Abstract

Stochastic Human Motion Prediction (HMP) has become an essential task for the realm of computer vision, for its capacity to anticipate accurate and diverse future human trajectories. Current diffusion-based techniques typically enforce skeletal consistency by encoding structural priors into network architectures. Although effective in promoting plausible kinematics, this approach provides only indirect control over the generative process and often fails to guarantee strict physical constraint satisfaction. In this work, we propose a structure-aligned and joint-aware diffusion framework that enforces physical constraints by embedding skeletal topology and joint-specific dynamics directly into the diffusion process. Specifically, our framework consists of two key modules, the Joint-Adaptive Noise Generator and the Structure-Aligned Regularizer. The former component, Joint-Adaptive Noise Generator, infers joint-specific dynamics and injects heterogeneous, instance-aware noise per joint and sample to capture spatial variability and enhance motion diversity. The latter component, Structure-Aligned Regularizer, encodes skeletal topology by modeling joint connectivity and bone lengths from historical motions, and it constrains each denoising step to preserve anatomical consistency. Through their synergistic operation, these modules grant KinemaDiff direct control over physical realism and motion diversity, addressing the common limitations of indirect structural priors and uniform noise application. Extensive experiments on multiple benchmarks demonstrate the effectiveness of our method, attributable to tailoring the diffusion process through structural alignment and joint-adaptive noise modeling.

## 1 Introduction

Human Motion Prediction (HMP) (Barsoum et al., 2018) aims to forecast future human motion sequences based on past observations, which is crucial for applications like autonomous driving (Paden et al., 2016), assistive robotics (Gui et al., 2018), and virtual avatars. While early deterministic methods (Xu et al., 2023; Li et al., 2022; Ma et al., 2022) sought to predict a single most likely future, they fell short in capturing the inherent unpredictability of human actions. Therefore, accurately modeling the multimodal and physically plausible distribution of future motions emerges as a paramount yet daunting task. To address this challenge, stochastic methods have gained prominence, with denoising diffusion probabilistic models becoming the mainstream approach (Yuan & Kitani, 2020; Dang et al., 2022). These models demonstrate remarkable capabilities in generating diverse motion sequences by progressively refining random noise into coherent human poses, as Fig. 1.

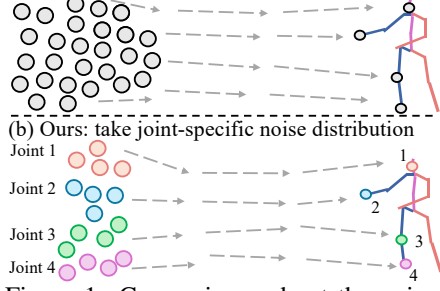

Figure 1: Comparisons about the noise taken between other baselines and ours.

Despite their generative power, these diffusion-based methods face two critical technical limitations in producing coherent and physically realistic human poses throughout the iterative diffusion pro-

---

*Equal contribution.

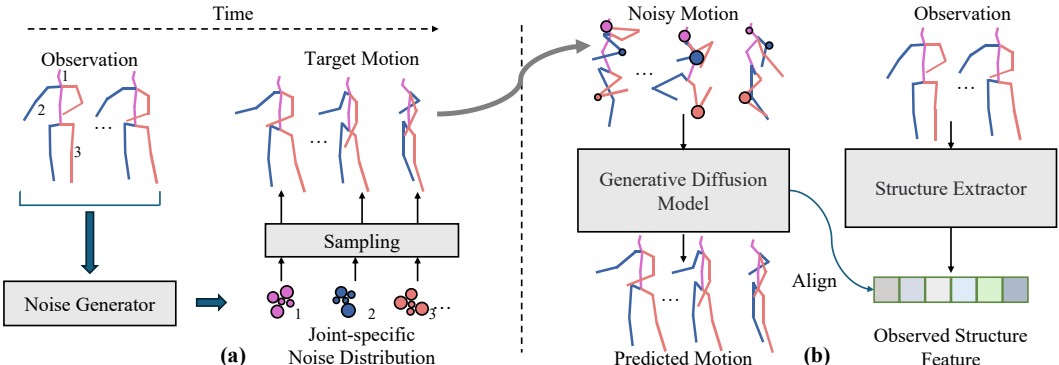

Figure 2: (a) Illustration of joint-adaptive noise generator. We propose a joint-adaptive noise determined by both the joint characteristics and the human motion observations. These noises are then added to the corresponding human motions to be predicted. (b) Representation of Structure-Aligned Regularizer, which identifies the human structural information from the historical motion and leverages the extracted structure to guide the motion generation during the diffusion process.

cess. On the one hand, a uniform noise schedule is typically applied across all human joints, failing to account for their heterogeneous motion patterns. Different joints exhibit vastly different degrees of freedom and dynamic behaviors. Applying identical noise profiles overlooks these unique kinematic properties, resulting in disordered or physically invalid predictions that compromise motion diversity and realism. On the other hand, prior methods tend to neglect the human skeleton's inherent anatomical structure. They often rely on implicitly learning the structural constraints (Chen et al., 2023; Sun & Chowdhary, 2024) or post-processing corrections (Wei et al., 2023), without integrating these constraints into the diffusion process, which leads to the generation of physically implausible poses with stretched or compressed bones, undermining motion realism, as in Fig. 2.

To address the aforementioned limitations, we present KinemaDiff, a novel kinematics-aware diffusion framework that fundamentally reshapes the denoising diffusion process, via explicitly embedding anatomical consistency and kinematic heterogeneity. Our framework consists of two core modules: the Joint-Adaptive Noise Generator and the Structure-Aligned Regularizer, as illustrated in Fig. 2. The first component, the Joint-Adaptive Noise Generator, is responsible for capturing and injecting heterogeneous motion patterns. It learns and applies instance-specific, heterogeneous noise profiles to different joints, effectively adapting the noise characteristics based on their unique dynamics and varied degrees of freedom, thereby guiding the diffusion process to generate dynamically rich and realistic motions. Subsequently, the Structure-Aligned Regularizer, is tasked with rigorously enforcing anatomical consistency throughout the generative process. It achieves this by directly integrating bone length constraints into the denoising procedure, leveraging stable structural features extracted from historical motion observations to ensure that generated poses adhere strictly to human biomechanics. Through the synergistic operation of these two modules, KinemaDiff enables direct and explicit control over physical realism and motion diversity, moving beyond the limitations of indirect structural priors and uniform noise application.

We extensively validate the effectiveness of our proposed diffusion model on Human3.6M and more challenging cross-dataset scenarios on AMASS. Our model outperformed previous models with multiple evaluation metrics on these datasets. Our contributions can be summarized as follows:

- We introduce KinemaDiff, a novel diffusion framework that integrates human skeletal structure and joint-specific motion dynamics directly within the diffusion process.
- We propose a learnable joint-adaptive noise generator to enhance motion diversity and a novel structural alignment mechanism to enforce anatomical consistency.
- We validate the effectiveness of our method through comprehensive experiments on Human3.6M and the cross-dataset AMASS benchmark.

## 2 RELATED WORK

**Stochastic Human Motion Prediction.** Early research in Human Motion Prediction (HMP) focused on deterministic forecasting using sequential models like RNNs and GCNs (Jain et al.,

2016; Dang et al., 2021). To capture the inherent multimodality of human actions, this field has shifted from deterministic forecasting to stochastic generation. Initial generative approaches, primarily Variational Autoencoders (VAEs) (Walker et al., 2017) and Generative Adversarial Networks (GANs) (Barsoum et al., 2018), pioneered the generation of diverse futures but often struggled with long-term coherence and physical plausibility. STARS (Xu et al., 2022) further advances multimodal motion modeling by introducing deterministic spatio-temporal anchors that enable diverse yet controllable future predictions. Recently, Denoising Diffusion Models (DMs) have become the dominant paradigm, offering superior fidelity and diversity. Research in this area has seen rapid progress, from pioneering the paradigm with a two-stage framework Motiondiff (Wei et al., 2023), simplifying the training pipeline via the masked completion model HumanMAC (Chen et al., 2023), to introducing latent diffusion for more coherent, behavior-driven sampling in BeLFusion (Barquero et al., 2023) and incorporating specialized architectures such as GCN-DCT to better capture spatio-temporal dynamics in CoMusion (Sun & Chowdhary, 2024). More recently, SLD (Xu et al., 2024) constructs a semantically structured latent motion space through Semantic Latent Directions, enabling precise and interpretable control over generated motions.

A concurrent line of work has also begun to adapt the diffusion process itself, such as Skeleton-Diffusion (Curreli et al., 2025), which introduces anisotropic noise based on the skeleton's static structure. In contrast to these approaches, which largely treat the core diffusion mechanism as a fixed component or adapt it based on static priors, our work is the first to fundamentally reshape the denoising process to be dynamically aware of human kinematics. We achieve this by introducing a novel framework equipped with a Structure-Aligned Regularizer for anatomical consistency and a learnable, instance-adaptive Joint-Adaptive Noise Generator, offering a more flexible and physically grounded generative process.

**Denoising Diffusion Probabilistic Models.** Denoising Diffusion Probabilistic Models (DDPMs) (Ho et al., 2020; Nichol & Dhariwal, 2021; Song et al., 2020) have emerged as a powerful class of generative models, capable of synthesizing high-fidelity data by learning to reverse a progressive noising process. While these models have been successfully applied to a wide range of human-centric tasks (Gong et al., 2023; Shan et al., 2023), such applications have predominantly focused on innovating the denoiser's network architecture, while largely adopting a standard, generic diffusion process. However, a recent and promising research direction has begun to demonstrate significant gains by tailoring the diffusion process itself, particularly through task-specific noise designs (Sahoo et al., 2024; Huang et al., 2024). Following this trajectory, we introduce a novel diffusion framework specifically for human motion, which moves beyond architectural modifications. We propose to fundamentally reshape the denoising process with a joint-adaptive, structure-aligned mechanism that is inherently adapted to the kinematic properties of human skeleton data.

# 3 METHODOLOGY

## 3.1 PROBLEM FORMULATION

As illustrated in Fig. 3, we denote the observed motion history of $H$ frames as $\mathbf{x}^{(1:H)} = [\mathbf{x}^{(1)}; \mathbf{x}^{(2)}; \ldots; \mathbf{x}^{(H)}] \in \mathbb{R}^{H \times 3J}$, where $\mathbf{x}^{(h)} \in \mathbb{R}^{3J}$ represents the joint coordinates at frame $h$, and $J$ is the total number of joints. Given $\mathbf{x}^{(1:H)}$, the goal of Human Motion Prediction is to forecast the subsequent $F$ frames $\mathbf{y}^{(1:F)} = \mathbf{x}^{(H+1:H+F)} = [\mathbf{x}^{(H+1)}; \mathbf{x}^{(H+2)}; \ldots; \mathbf{x}^{(H+F)}] \in \mathbb{R}^{F \times 3J}$, where $\mathbf{y}^{(f)} \in \mathbb{R}^{J \times 3}$, and $J$ is the number of body joints.

## 3.2 PRELIMINARIES

**Motion Diffusion.** Let $\{y_t\}_{t=0}^{T}$ denote a Markov noising process, where $y_0$ represents the true data samples. For training, we progressively corrupt the target human motion sequence $\{y_t\}_{t=0}^{T}$ by adding noise. This forward diffusion process is represented as:

$$q(y_t \mid y_{t-1}) = \mathcal{N}\big(y_t; \sqrt{\alpha_t}\, y_{t-1},\, (1 - \alpha_t)\mathbf{I}\big), \tag{1}$$

where $\{\alpha_t\}_{t=0}^{T} \in [0, 1]$ controls the noise level. To approximate the underlying data distribution, the reverse diffusion process is formulated to iteratively remove noise from the corrupted samples $y_t$, starting from $t = T$ down to $t = 1$, as follows:

$$p_\theta(y_{t-1} \mid y_t) = \mathcal{N}\big(y_{t-1}; \mu_\theta(y_t, t), \sigma_\theta^2(y_t, t)\mathbf{I}\big). \tag{2}$$

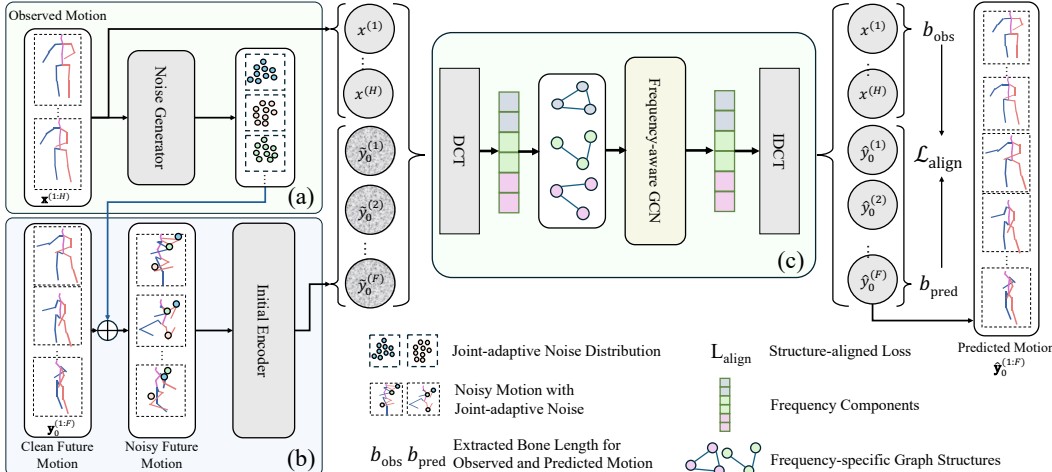

Figure 3: The overview of our proposed Kinemadiff. (a): Our Joint-adaptive noise generator. We learn joint-adaptive noise from the historical human joints and add it to the future human motions. (b): Initial motion reconstruction. The future human motion with injected noise is processed through a self-attention mechanism, which generates an initial prediction in the absence of external conditioning. (c): Structure-Aligned Regularizer. The initial prediction is concatenated with the motion observations and then processed in the frequency domain through a frequency-aware GCN, and subsequently transformed back to the temporal domain for structural alignment.

In addition, following prior work (Chen et al., 2023; Barquero et al., 2023), we condition the model on the historical motion. The conditional reverse diffusion transition is then formulated as:

$$p_\theta(y_{t-1} \mid y_t, x) = \mathcal{N}\big(y_{t-1}; \mu_\theta(y_t, x, t), \sigma_\theta^2(y_t, x, t)\mathbf{I}\big). \tag{3}$$

where $x$ denotes the motion history and $y_t$ the noisy motion at step $t$.

**Direct $\mathbf{y}_0$ prediction.** In existing diffusion models (Chen et al., 2023; Barquero et al., 2023), the objective is typically either to predict noise or to directly predict human motion. In this work, we adopt the latter, as it enables more effective optimization of the diffusion process through the incorporation of skeletal structural priors.

### 3.3 OVERALL ARCHITECTURE AND NETWORK DESIGN

**Overall Architecture.** As illustrated in Fig. 3, our network takes the observed motion history $\mathbf{x}^{(1:H)} \in \mathbb{R}^{H \times 3J}$ and the future frames $\mathbf{y}^{(1:F)} \in \mathbb{R}^{F \times 3J}$ as input. The future frames are perturbed with our joint-adaptive noise, which varies across joints and samples, yielding a sequence $\{y_t\}_{t=0}^{T}$ that follows the forward diffusion process. Our model is trained to reverse this process and ultimately reconstruct the predicted future motion $\hat{\mathbf{y}}_0 \in \mathbb{R}^{F \times 3J}$. First, we feed the noisy future frames $\mathbf{y}_t$ into an encoder composed of several Transformer layers to obtain an initial reconstruction $\tilde{\mathbf{y}}_0 \in \mathbb{R}^{F \times 3J}$. To ensure structural alignment, we reshape the motion history as $\mathbf{x}^{(1:H)} \in \mathbb{R}^{H \times J \times 3}$ and the noisy future as $\tilde{\mathbf{y}}_0 \in \mathbb{R}^{F \times J \times 3}$. Next, we concatenate the initial reconstruction with the historical motion and process it with the Alignment Module to produce the denoised prediction.

**Network Structure.** As illustrated in Fig. 3, our network architecture consists of three main components: a joint-adaptive noise generator, an initial encoder, and a Structure-Aligned Regularizer. The first component generates joint-adaptive noise conditioned on a fixed-length motion history and joint characteristics using a few linear layers. The second component employs a temporal Transformer encoder to generate an initial reconstruction, providing a baseline prediction without conditioning. The third component is the proposed Structure-Aligned Regularizer, which models the full motion sequence in the frequency domain via DCT/IDCT. Unlike prior works, it employs GCNs with frequency-specific adjacency matrices, where each frequency band is associated with a tailored connectivity to capture its distinct motion patterns, thereby enabling more effective modeling of motion dynamics across different bands.

### 3.4 JOINT-ADAPTIVE NOISE GENERATOR.

Unlike previous diffusion-based methods that adopt a fixed schedule, our approach designs a learnable noise schedule. We introduce a multivariate noise schedule that assigns joint-adaptive noise rates, enabling the diffusion process to capture spatial variability in the human skeleton:

$$q(y_t \mid y_{t-1}) = \mathcal{N}\big(y_t; \alpha_t y_{t-1}, (1 - \alpha_t)\Sigma\big) \tag{4}$$

where $\Sigma = \mathrm{diag}(s_1^2, s_2^2, \ldots, s_J^2)$, $s_j$ denotes the noise scaling factor for the $j$-th joint. Specifically, the scaling factor is determined by two aspects. First, it depends on the joint index, since different joints exhibit distinct motion characteristics. Second, we further refine the scaling factor by conditioning on the historical motion trajectories of each joint. Formally, the joint- and instance-specific noise scaling can be expressed as:

$$s_j = f_\theta(j, \mathbf{x}_j^{(1:H)}) \tag{5}$$

where $j$ is the joint index, $\mathbf{x}_j^{(1:H)}$ denotes the observed motion history of joint $j$, and $f_\theta$ is a learnable function. Through this design, we are able to inject noise with varying intensities across different joints, better reflecting their heterogeneous motion properties. The reverse diffusion process that denoises the corrupted data samples $y_t$ from $t = T$ down to $t = 1$:

$$p_\theta(y_{t-1} \mid y_t) = \mathcal{N}\big(y_{t-1}; \mu_\theta(y_t, x, t), (1 - \alpha_t)\Sigma\big) \tag{6}$$

where $\Sigma = \mathrm{diag}(s_1^2, s_2^2, \ldots, s_J^2)$, $s_j$ denotes the noise scaling factor for the $j$-th joint. Through the above design, the diffusion process is able to inject joint- and instance-adaptive noise, leading to more realistic and coherent human motion generation.

### 3.5 STRUCTURE-ALIGNED REGULARIZER

Earlier diffusion-based approaches (Chen et al., 2023; Sun & Chowdhary, 2024) treated human motion prediction as a straightforward application of diffusion, overlooking the intrinsic structural properties of human pose. For example, such methods fail to enforce structural constraints, such as maintaining consistent bone lengths throughout motion generation. We proposed a Structure-Aligned Regularizer, which effectively aligns the predicted human motion with historical motion patterns based on the human skeletal structure during the diffusion process. Specifically, we calculated the average bone length of connected joints from past human motions. Since the historical motion is noise-free, it allows us to easily extract structural information. We regard the motion to be predicted as clean human motion plus Gaussian noise:

$$y_t = \sqrt{\bar{\alpha}_t}\, y_0 + \sqrt{1 - \bar{\alpha}_t}\, \epsilon, \tag{7}$$

where $\epsilon \sim \mathcal{N}(0, I)$ denotes Gaussian noise and $y_0$ is the clean motion. For a batch of $y_t$, we take the mean across the batch. Since the noise $\epsilon$ follows a Gaussian distribution with zero mean, we can compute the bone lengths of the motion sequence $\mathbf{y}$ from this mean:

$$\bar{y}_t = \frac{1}{B} \sum_{b=1}^{B} y_t^{(b)} \approx \sqrt{\bar{\alpha}_t}\, y_0, \tag{8}$$

where $B$ is the batch size. Specifically, we denote the set of skeletal connections as $\mathcal{E}$, where each $(i, j) \in \mathcal{E}$ represents a connected joint pair. The corresponding bone length is defined as:

$$\ell_{i,j} = \|y^i - y^j\|_2, \quad (i, j) \in \mathcal{E}, \tag{9}$$

where $y^i, y^j \in \mathbb{R}^3$ denote the 3D coordinates of joints $i$ and $j$.

As previously mentioned, our model predicts the future motion values directly at each step. When $t$ is relatively large, the direct prediction of $\mathbf{y}_0$ tends to be inaccurate. To address this, we impose constraints on the bone lengths of the target human motion and perform an alignment with the structure of the historical human motion. Moreover, at each timestep, after the initial encoder, we apply the same operation on $\tilde{\mathbf{y}}_0$ to ensure that the human skeleton structure remains consistent throughout the entire sequence. Specifically, for each bone connection $(i, j) \in \mathcal{E}$, we compute the average bone length over the observed history frames $\mathbf{x}^{(1:H)}$ as $\bar{b}_{\mathrm{obs}}^{(i,j)}$, and those of $\hat{\mathbf{y}}_0^{(1:F)}$ and $\tilde{\mathbf{y}}_0^{(1:F)}$

as $\bar{b}_{\text{pred}}^{(i,j)}$ and $\bar{b}_{\text{ref}}^{(i,j)}$, respectively. The alignment loss is then defined as the mean discrepancy between the two sets of averaged bone lengths:

$$\mathcal{L}_{\text{align}} = \frac{1}{|\mathcal{E}|} \sum_{(i,j) \in \mathcal{E}} \left| \bar{b}_{\text{obs}}^{(i,j)} - \bar{b}_{\text{pred}}^{(i,j)} \right|_2 + \frac{1}{|\mathcal{E}|} \sum_{(i,j) \in \mathcal{E}} \left| \bar{b}_{\text{obs}}^{(i,j)} - \bar{b}_{\text{ref}}^{(i,j)} \right|_2, \tag{10}$$

which encourages the predicted skeleton to preserve the structural scale of the observed motion, thereby maintaining consistent bone proportions across time.

## 3.6 OVERALL LEARNING OBJECTIVES

Our loss consists of two components: a reconstruction loss applied to the predicted poses and an alignment loss enforcing consistency between the predicted human motion and observed human motion. Unlike most diffusion-based motion models that predict noise at each timestep, our denoiser directly outputs a pose prediction $\hat{y}_0$ for every diffusion timestep $t$. This design allows us to impose the reconstruction and alignment loss at every timestep, ensuring that the denoising trajectory remains consistently aligned with anatomically plausible human motion rather than relying solely on the final-step supervision. For the reconstruction loss, we follow prior work (Sun & Chowdhary, 2024) and assign different weights to individual joints, which are weighted differently to reflect their relative importance in the motion context. The reconstruction loss for each timestep is defined by:

$$\mathcal{L}_{\text{rec}} = \frac{1}{J} \sum_{j=1}^{J} \left( \gamma \cdot \left\| (x^j - \hat{x}^j) \cdot \lambda^j \right\|_1 + \left\| (y_0^j - \hat{y}_0^j) \cdot \lambda^j \right\|_1 \right), \tag{11}$$

where the superscript $j$ denotes the joint index, $\lambda^j$ is the weight assigned to each joint, and $\gamma$ is a hyperparameter balancing the reconstruction of motion history and the prediction of future.

$$\mathcal{L}_{\text{total}} = \alpha \cdot \mathcal{L}_{\text{rec}} + \beta \cdot \mathcal{L}_{\text{align}}, \tag{12}$$

where $\alpha$ and $\beta$ control the relative weight of the reconstruction and alignment losses.

## 4 EXPERIMENTS

We first introduce the experimental setup in §4.1. Then we assess the performance of our method across various settings, including intra-dataset forecasting on Human3.6M(§4.2), and more challenging cross-dataset generalization on AMASS(§4.3). Lastly, we provide ablative analyses in §4.4.

## 4.1 EXPERIMENTAL SETUP

**Datasets.** We conduct experiments on two widely used datasets, intra-dataset forecasting on Human3.6M (Ionescu et al., 2013) and cross-dataset generalization on AMASS (Mahmood et al., 2019).

- **Human3.6M** is a seminal indoor dataset for 3D human motion analysis, widely utilized in stochastic Human Motion Prediction. It comprises 3.6 million frames, captured at 50Hz, documenting 11 subjects performing 15 common daily activities. Consistent with prior work (Barquero et al., 2023), we delineate subjects S1, S5, S6, S7, S8 for training, and subjects S9, S11 for evaluation.
- **AMASS** is a large-scale, highly diverse motion dataset for assessing cross-dataset generalization. It consolidates 24 distinct motion capture datasets, all standardized to the SMPL parameterization, accumulating over 9 million frames. Following prior research (Barquero et al., 2023), the dataset is partitioned into 11 training, 4 validation, and 7 testing constituent datasets.

**Implementation Details.** Our model, is trained end-to-end, following protocols as detailed below:

- **Diffusion Settings.** KinemaDiff employs a 10-step diffusion process with standard DDPM sampling, which is inherently augmented by our proposed kinematics-aware designs. Specifically, both the Structure-Aligned Regularizer and the Joint-Adaptive Noise Generator are integrated within the diffusion steps to ensure physical consistency and capture joint heterogeneity.

Table 1: Quantitative results on Human3.6M. The best results are highlighted in **bold**. The symbol '–' indicates not reported in the baseline work. For all metrics except for APD, lower is better.

| Method | Reference | Accuracy | | Multimodality | | Diversity | Realism | |
| --- | --- | --- | --- | --- | --- | --- | --- | --- |
| | | ADE ↓ | FDE ↓ | MMADE ↓ | MMFDE ↓ | APD ↑ | CMD ↓ | FID ↓ |
| *GAN-Based* | | | | | | | | |
| HP-GAN (Barsoum et al., 2018) | [CVPRW2018] | 0.858 | 0.867 | 0.847 | 0.858 | 7.214 | – | – |
| DeLiGAN (Gurumurthy et al., 2017) | [CVPR2017] | 0.483 | 0.534 | 0.520 | 0.545 | 6.509 | – | – |
| *VAE-Based* | | | | | | | | |
| TPK (Walker et al., 2017) | [ICCV2017] | 0.461 | 0.560 | 0.522 | 0.569 | 6.723 | 6.326 | 0.538 |
| Motron (Salzmann et al., 2022) | [CVPR2022] | 0.375 | 0.488 | 0.509 | 0.539 | 7.168 | 40.796 | 13.743 |
| DSF (Yuan & Kitani, 2019) | [ICLR2020] | 0.493 | 0.592 | 0.550 | 0.599 | 9.330 | – | – |
| DLow (Yuan & Kitani, 2020) | [ECCV2020] | 0.425 | 0.518 | 0.495 | 0.531 | 11.741 | 4.927 | 1.255 |
| GSPS (Mao et al., 2021) | [ICCV2021] | 0.389 | 0.496 | 0.476 | 0.525 | 14.757 | 10.758 | 2.103 |
| DivSamp (Dang et al., 2022) | [ACMMM2022] | 0.370 | 0.485 | 0.475 | 0.516 | 15.310 | 11.692 | 2.083 |
| *DM-Based* | | | | | | | | |
| MotionDiff (Wei et al., 2023) | [AAAI2023] | 0.411 | 0.509 | 0.508 | 0.536 | **15.353** | – | – |
| HumanMAC (Chen et al., 2023) | [ICCV2023] | 0.369 | 0.480 | 0.509 | 0.545 | 6.301 | – | – |
| BeLFusion (Barquero et al., 2023) | [ICCV2023] | 0.372 | 0.474 | **0.473** | 0.507 | 7.602 | 5.988 | 0.209 |
| CoMusion (Sun & Chowdhary, 2024) | [ECCV2024] | 0.350 | 0.458 | 0.494 | **0.506** | 7.632 | 3.202 | 0.102 |
| SkeletonDiff (Curreli et al., 2025) | [CVPR2025] | 0.344 | 0.450 | 0.487 | 0.512 | 7.249 | 4.178 | 0.123 |
| Ours | - | **0.331** | **0.449** | 0.500 | 0.520 | 6.912 | 4.60 | **0.083** |

- **Training Protocols.** For both the Human3.6M and AMASS datasets, the model undergoes training for 500 epochs. We utilize the AdamW optimizer with a batch size of 128. The initial learning rate is set to 1e-4, which is subsequently decayed after the 200th epoch. These training parameters are consistent across both datasets.
- **Dataset-Specific Protocols.**
  - **Human3.6M.** Consistent with prior work (Barquero et al., 2023), we predict 100 future frames from 25 observed frames, using a 16-joint skeleton.
  - **AMASS.** Following the protocol established by (Barquero et al., 2023), the task involves forecasting 120 future frames (2s) based on 30 observed frames (0.5s).

**Baselines.** We compare our method against several representative diffusion-based methods.

- **BeLFusion**. Operating within a VAE-encoded latent space, BeLFusion employs a diffusion model to generate diverse and behavior-driven future motion predictions. Nevertheless, its physics-based consistency check is a post-processing step applied externally, rather than a constraint that guides the iterative denoising process internally.
- **CoMusion**. Integrating Graph Convolutional Networks (GCNs) within the Discrete Cosine Transform (DCT) space, CoMusion's denoiser effectively captures complex spatio-temporal dependencies. Though effective in modeling these dependencies, it focuses on advancing the architecture, rather than proposing adaptations to the core diffusion mechanism itself.
- **SkeletonDiffusion**. Introducing a non-isotropic diffusion process, SkeletonDiffusion defines an anisotropic noise covariance matrix derived from the skeleton's static kinematic tree. Though effective in acknowledging joint heterogeneity, its noise characteristics remain fixed and are not adaptive to the unique dynamics of a given motion instance.

**Evaluation Metrics.** Following established practices, we evaluate our method across three key aspects: accuracy, diversity, and realism, using a comprehensive suite of standard metrics.

- **Accuracy.** We report Average Displacement Error (ADE) and Final Displacement Error (FDE), the mean $\ell_2$ distance to the GT over the sequence and at the final frame, respectively.
- **Diversity and Multimodality.** We take Average Pairwise Distance (APD) to measure the variance among generated samples. we also report Multimodal ADE/FDE (MMADE/MMFDE), which assess multimodality by measuring the error to the best-matching ground-truth variant.
- **Realism and Plausibility.** We employ the Fréchet Inception Distance (FID) to assess the distributional similarity between generated and real motions. Additionally, following (Barquero et al.,

Table 2: Quantitative results for AMASS dataset. The best results are highlighted in **bold**. As AMASS does not contain class labels, the FID metric is not used for evaluation.

| Method | Reference | Accuracy | | Multimodality | | Diversity | Realism |
|---|---|---|---|---|---|---|---|
| | | ADE ↓ | FDE ↓ | MMADE ↓ | MMFDE ↓ | APD ↑ | CMD ↓ |
| *VAE-Based* | | | | | | | |
| TPK (Walker et al., 2017) | [ICCV2017] | 0.656 | 0.675 | 0.658 | 0.674 | 9.283 | 17.127 |
| DLow (Yuan & Kitani, 2020) | [ECCV2020] | 0.590 | 0.612 | 0.618 | 0.617 | 13.170 | 15.185 |
| GSPS (Mao et al., 2021) | [ICCV2021] | 0.563 | 0.613 | 0.609 | 0.633 | 12.465 | 18.404 |
| DivSampp (Dang et al., 2022) | [ACMMM2022] | 0.564 | 0.647 | 0.623 | 0.667 | **24.724** | 50.239 |
| *DM-Based* | | | | | | | |
| HumanMAC (Barquero et al., 2023) | [ICCV2023] | 0.511 | 0.554 | 0.593 | 0.591 | 9.321 | – |
| BeLFusion (Barquero et al., 2023) | [ICCV2023] | 0.513 | 0.560 | 0.569 | 0.585 | 9.376 | 16.995 |
| CoMusion (Sun & Chowdhary, 2024) | [ECCV2024] | 0.494 | 0.547 | 0.469 | 0.466 | 10.848 | 9.636 |
| SkeletonDiff (Curreli et al., 2025) | [CVPR2025] | 0.480 | 0.545 | 0.561 | 0.580 | 9.456 | 11.417 |
| Ours | - | **0.478** | **0.540** | **0.456** | **0.457** | 9.683 | **9.448** |

Table 3: Ablation of the main components in our method on Human3.6M.

| Encoder | J-Noise | Align | APD ↑ | ADE ↓ | FDE ↓ | FID ↓ |
|---|---|---|---|---|---|---|
| - | - | - | **19.601** | 0.852 | 0.775 | 2.393 |
| ✓ | - | - | 9.600 | 0.653 | 0.574 | 0.932 |
| - | - | ✓ | 6.214 | 0.354 | 0.478 | 0.177 |
| ✓ | - | ✓ | 7.243 | 0.339 | 0.454 | 0.088 |
| ✓ | ✓ | - | 7.014 | 0.336 | 0.453 | 0.089 |
| ✓ | ✓ | ✓ | 6.912 | **0.331** | **0.449** | **0.083** |

Table 4: Experiment results on Human3.6M with different Scheduler.

| Scheduler | APD ↑ | ADE ↓ | FDE ↓ | FID ↓ |
|---|---|---|---|---|
| Sqrt | 6.837 | 0.342 | 0.457 | 0.108 |
| Cosine | 7.213 | 0.365 | 0.478 | 0.178 |
| Variance | 6.912 | 0.331 | 0.449 | 0.083 |

2023), we use the Cumulative Motion Distribution (CMD) area for global plausibility to evaluate how realistically the model's generated diversity reflects that of the ground truth.

## 4.2 RESULTS OF INTRA-DATASET FORECASTING ON HUMAN3.6M

The comparative performance on the Human3.6M dataset is systematically reported in Tab. 1. The results unequivocally demonstrate that our model establishes a new state-of-the-art result in forecasting accuracy and realism. Specifically, KinemaDiff achieves an ADE of **0.331** and an FDE of **0.449**, surpassing all prior methods. We attribute this superior accuracy to the Structure-Aligned Regularizer. By rigorously maintaining anatomical consistency at each step of the denoising process, this module prevents the accumulation of kinematic errors that can degrade long-term predictions, ensuring a physically grounded and accurate motion trajectory.

Furthermore, the effectiveness of our approach in generating high-fidelity motion is underscored by the FID score of **0.083**, a substantial 19% relative improvement over the previous leading model, CoMusion. A lower FID indicates that the distribution of generated motions is significantly closer to that of real human movements, not just in individual poses but in the naturalness of the entire sequence. This gain in realism is a direct result of the synergy between our two core modules: the Structure-Aligned Regularizer eliminates anatomically impossible poses, while the Joint-Adaptive Noise Generator sculpts more natural, heterogeneous joint movements, avoiding the robotic uniformity that can arise from conventional noise schedules. While other methods may achieve higher raw diversity scores (APD) or broader multimodal coverage (MMADE/FDE), KinemaDiff excels at ensuring that every generated sample possesses high physical fidelity, prioritizing the quality and plausibility of predictions as evidenced by its leading accuracy and realism metrics.

## 4.3 RESULTS OF CROSS-DATASET GENERALIZATION ON AMASS

To evaluate robustness and generalization, we present a comparative analysis on the diverse AMASS dataset in Tab. 2. In this challenging cross-dataset scenario, KinemaDiff demonstrates exceptional performance, achieving advanced results across the majority of metrics, including **ADE (0.478)**, **FDE (0.540)**, **MMADE (0.456)**, **MMFDE (0.457)**, and **CMD (9.448)**. These results highlight

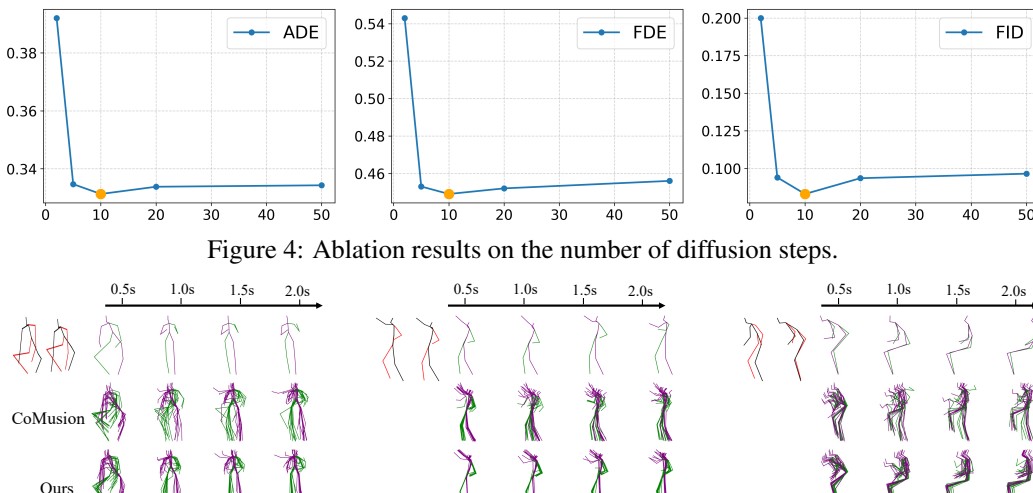

Figure 4: Ablation results on the number of diffusion steps.

Figure 5: Visualization results. The red-black skeletons and green-purple skeletons denote the observed and predicted motions respectively.

the model's ability to learn fundamental principles of motion rather than dataset-specific artifacts. The superior generalization is primarily driven by the Structure-Aligned Regularizer, which learns intrinsic and invariant anatomical properties like bone lengths, making the model robust to the wide variety of novel motions present in AMASS. Concurrently, the results in multimodal metrics (MMADE/MMFDE) provide clear evidence for the efficacy of the Joint-Adaptive Noise Generator. On a diverse dataset like AMASS, where a single observation can lead to many valid future actions, the ability to generate instance-specific, heterogeneous noise allows the model to explore this rich possibility space more effectively than methods with static or uniform noise. It does not merely generate random variations but rather meaningful and plausible alternatives tailored to the input context. While some methods achieve a higher raw diversity score (APD) at the cost of accuracy, KinemaDiff achieves a competitive APD (9.683) while simultaneously delivering the best accuracy and multimodal coverage, demonstrating a superior balance between diversity and fidelity.

## 4.4 ABLATION STUDY

For in-depth analysis, we conduct ablative studies using Intra-Dataset Forecasting on Human3.6M.

**Effect of our main components.** To assess the contribution of our core components, we conducted an ablation study by removing the Structure-Aligned Regularizer, joint-adaptive noise, and the initial encoder. From Tab. 3, it can be seen that using only the baseline or only the initial encoder leads to low accuracy (ADE, FDE) and FID scores, despite high APD values. This suggests the model produces motions that are diverse yet largely implausible and unstructured. Incorporating the Structure-Aligned Regularizer substantially improves accuracy metrics (ADE, FDE) and consistency metrics (FID) by leveraging structural cues aligned with historical motion. In addition, the joint-adaptive noise dynamically allocates noise to different joints, enabling more accurate modeling of human motion and further improving both accuracy (ADE, FDE) and consistency (FID) metrics.

**Diffusion setting.** We mainly investigate two aspects of the diffusion setting: the choice of scheduler and the number of diffusion steps. We evaluate multiple scheduler choices on the Human3.6M dataset in Tab. 4, and the results indicate that Variance Scheduler (Sun & Chowdhary, 2024) provides the best trade-off between stability and performance. In addition, we investigate the effect of the denoising step on model performance. We evaluate multiple metrics on Human3.6M under different timesteps. As shown in Fig. 4, the model achieves the best performance when the number of timesteps is 10. Moreover, choosing 10 timesteps ensures fast inference and accuracy.

**Visualization results.** In Fig. 5, we present a qualitative comparison by visualizing predicted motion sequences on Human3.6M. We use CoMusion as the baseline and select 15 predicted results for each frame. The visual analysis shows that our method generates more consistent and realistic human motions. Compared with the ground truth, our predictions are especially accurate for samples with smooth movements. In addition, unrealistic artifacts, such as sudden exaggerated leg lifts while

Table 5: Comparison of physical realism metrics on Human3.6M.

| Method | Limb Stretch ↓ | Limb Jitter ↓ | ADE ↓ | FDE ↓ | FID ↓ |
|---|---|---|---|---|---|
| SkeletonDiffusion | 3.90 | **0.16** | 0.344 | 0.450 | 0.123 |
| KinemaDiff (Ours) | **2.42** | 0.45 | **0.331** | 0.449 | **0.083** |
| KinemaDiff + jitter loss | **2.42** | 0.28 | **0.331** | **0.447** | 0.084 |

walking, occur less frequently. Moreover, the diversity of generated motions better reflects realistic dynamics, with predictions concentrated near historical patterns and gradually diffusing over time.

**Analysis of Physical Realism Metrics.** To quantitatively assess the physical plausibility of the generated motions, we evaluate two additional metrics on Human3.6M: Limb Stretch and Limb Jitter, as seen in Tab. 5. Our method achieves a limb stretch of 2.42, which is lower than SkeletonDiffusion's 3.90. This improvement is primarily attributed to our Structure-Aligned Regularizer, which constrains the skeleton at every diffusion timestep, effectively reducing unrealistic deformations. Besides, our model obtains a limb jitter score of 0.45. To further explore the flexibility of our framework in enforcing temporal consistency, we adopted the same jitter loss as used in SkeletonDiffusion as an additional constraint. Specifically, this loss quantifies the magnitude of bone length changes between adjacent frames. We calculate the bone length $L_t^j$ from 3D keypoints and define the jitter as the absolute difference $|L_{t+1}^j - L_t^j|$. The loss minimizes the squared difference between the predicted and ground-truth jitter values, constraining the generated bone dynamics to be temporally consistent with real motion. After incorporating this loss, the jitter metric significantly improves from 0.45 to 0.28. This demonstrates that our framework can readily incorporate such constraints to prevent excessive high-frequency shaking. Crucially, adding this loss does not compromise other metrics (ADE / FDE / FID remain stable), confirming that our framework can effectively balance structural validity and temporal smoothness.

**Effect of different loss functions in Structure-aligned Regularizer.** To further investigate the impact of different structural regularization strategies, we evaluate two stricter frame-wise bone-length constraints based on $L1$ and $L2$ penalties, as seen in Tab. 6. The $L1$ formulation leads to substantially degraded performance (ADE 0.457), indicating that its constant gradient may introduce instability during denoising. In contrast, the $L2$-based constraint enforces the geometry more rigorously and achieves the lowest FID, but slightly reduces prediction accuracy due to over-constraining the motion manifold at each timestep.

Table 6: Effect of different loss functions in Structure-Aligned Regularizer.

| Loss Variant | ADE ↓ | FDE ↓ | FID ↓ |
|---|---|---|---|
| Frame-wise $L_1$ | 0.457 | 0.574 | 0.512 |
| Frame-wise $L_2$ | 0.333 | 0.452 | **0.075** |
| Average (Ours) | **0.331** | **0.449** | 0.083 |

## 5 CONCLUSION

In this work, we introduce a new diffusion model tailored for stochastic human motion prediction. Our algorithm integrates a Joint-Adaptive Noise Generator and a Structure-Aligned Regularizer directly within the diffusion process. The former enhances motion diversity with instance-aware noise , while the latter preserves anatomical consistency by embedding structural priors into each diffusion step. Results across multiple benchmarks demonstrate its effectiveness, owing to the unique integration of structural and dynamic priors within diffusion process.

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

# A   APPENDIX

The supplementary material herein extends the discussion and analysis presented in the primary manuscript. It is structured as follows:

**Ethics statement.** (§ A.1) : This section describes ethics statement of the manuscript.

**Reproducibility Statement.** (§ A.2) : This section describes reproducibility statement of the manuscript.

**Use of Large Language Models** (§ A.3) : This section describes the use of large language models across the manuscript.

**Additional Visualization** (§ A.4) : This section introduces additional visualization results.

**Additional Hyperparameter Setting** (§ A.5) : This section introduces additional hyperparameter setting.

**Additional Metric Descriptions** (§ A.6) : This section introduces the details of the metric descriptions used in the manuscript.

**Comparison with anisotropic noise** (§ A.7) : This section introduces a comparison with anisotropic noise and more detailed effect of the Joint-Adaptive Noise Generator.

**Comparison with Standard Bone-Length Constraints** (§ A.8) : This section introduces the comparison with standard bone-length constraints.

**Discussion of learned noise scale** (§ A.9): This section discusses the learned noise scale.

**Discussion on realism/accuracy and diversity of our model** (§ A.10): This section analyzes how enforcing anatomical consistency leads to high-quality, physically plausible motion diversity, even if numerical diversity (APD) is slightly reduced.

**Effectiveness of Early-Timestep Structural Constraint** (§ A.11): This section demonstrates why applying structural constraints at early timesteps is effective.

## A.1   ETHICS STATEMENT.

This work focuses on human motion prediction using publicly available benchmark datasets (Human3.6M and AMASS), which were collected and released under established research protocols. No personally identifiable or sensitive information is involved, and our methodology does not involve human subjects, sensitive attributes, or private data, posing no privacy or security concerns. All experiments follow ethical research practices, including proper citations, fair comparisons with prior works, and reproducibility efforts. All authors have read and will adhere to the ICLR Code of Ethics.

## A.2   REPRODUCIBILITY STATEMENT.

We have made significant efforts to ensure the reproducibility of our work. A detailed description of our model architecture is provided in Section 3, while the evaluation protocols and training setup are presented in Section 4 of the main paper. Additional implementation details are included in the Appendix. All datasets used in our experiments are publicly available and widely adopted benchmarks, and our preprocessing steps strictly follow prior works to ensure consistency and comparability.

## A.3   THE USE OF LARGE LANGUAGE MODELS (LLMS)

In preparing this manuscript, we used a large language model (LLM) only for grammar correction and improving the clarity of phrasing. The LLM was not involved in generating ideas, methods, experiments, analyses, or results. All scientific contributions, including the problem formulation, model design, and evaluation, are entirely the work of the authors.

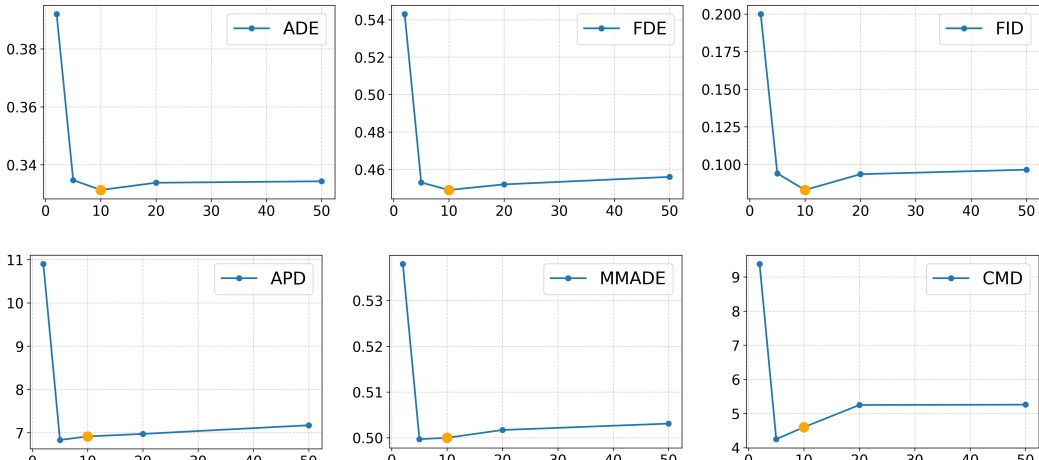

Figure 6: Additional ablation results on the number of diffusion steps.

## A.4 ADDITIONAL VISUALIZATION.

To further validate the physical realism and accuracy of our generated motions, we provide additional qualitative comparisons with baselines including CoMusion (Sun & Chowdhary, 2024), SkeletonDiffusion (Curreli et al., 2025), and BeLFusion (Barquero et al., 2023). As illustrated in the accompanying figures (Fig. 7), KinemaDiff consistently produces results closer to the ground truth with fewer unrealistic poses compared to CoMusion. Moreover, it significantly reduces artifacts like bone stretching common in SkeletonDiffusion and BeLFusion. These visual improvements substantiate the effectiveness of our approach, demonstrating that our proposed structural constraints and joint-adaptive noise generator effectively enforce anatomical consistency and improve motion coherence without compromising diversity.

## A.5 ADDITIONAL HYPERPARAMETER SETTING.

**Additional diffusion setting.** We introduce three additional metrics—APD, MMADE, and CMD—to evaluate model performance across different timesteps. As shown in Fig. 6, when the number of timesteps is set to 10, the model achieves strong performance in terms of diversity, accuracy, and consistency. Therefore, we select this setting for our experiments.

**Hyperparameter for loss function.** In Section 3.6, we set $\gamma = 0.1$ to balance the reconstruction of motion history and the prediction of future frames. The coefficients $\alpha$ and $\beta$ are set to 1 and 2, respectively, to control the relative contributions of the reconstruction loss and the alignment loss. These values were selected based on experiments with several parameter configurations to identify the most effective setting.

**Model setting.** In the initial encoder, we stack two Transformer layers with a feature dimension of 512. In the Structure-Aligned Regularizer, we employ nine Frequency-aware GCN layers with a feature dimension of 125. The graph structure consists of $N$ nodes, where $N$ corresponds to the number of joints in the skeleton.

## A.6 ADDITIONAL METRIC DESCRIPTIONS.

- **APD (Average Pairwise Distance)** measures the diversity of generated samples by computing the average distance between all pairs of generated motions.
- **ADE (Average Displacement Error)** computes the mean per-timestep distance between the predicted and ground-truth motions, reflecting overall accuracy.
- **FDE (Final Displacement Error)** measures the distance between the predicted and ground-truth motions at the final timestep, highlighting long-term prediction accuracy.
- **MMADE and MMFDE** extend ADE/FDE by comparing with clustered groundtruth variants, capturing a model's ability to generate multiple plausible outcomes.

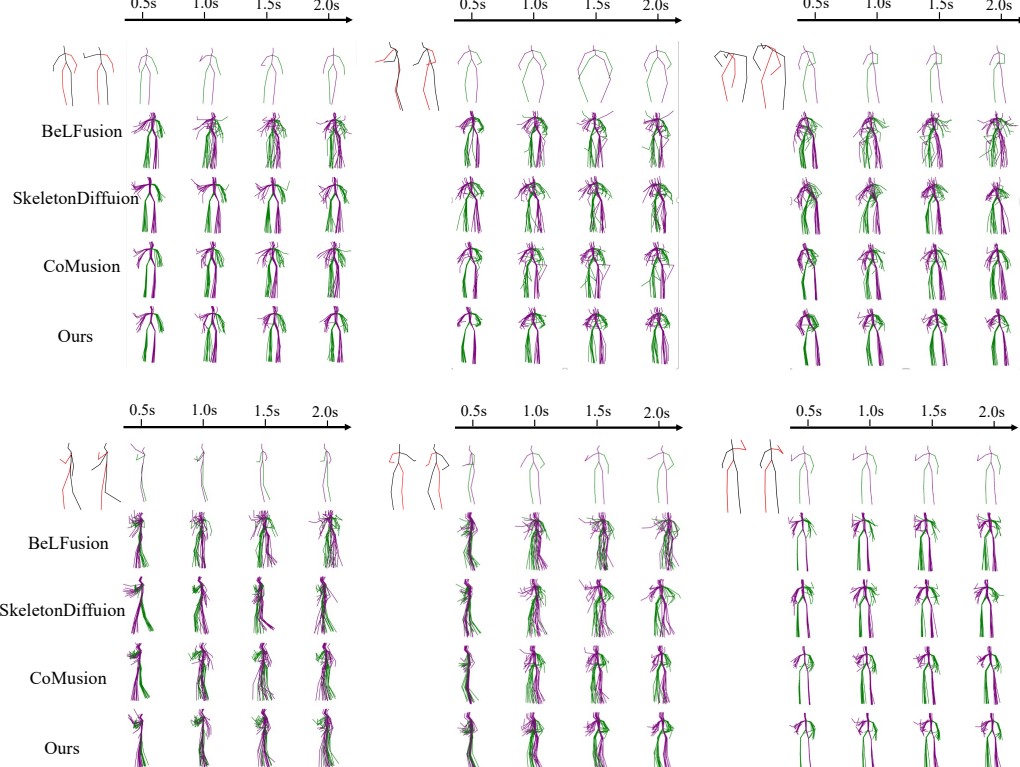

Figure 7: Additional visualization results. The red-black skeletons and green-purple skeletons denote the observed and predicted motions respectively.

- **CMD (Conditional Motion Distance)** quantifies global plausibility by comparing the areas under cumulative distributions of true and generated motion.
- **FID (Fréchet Inception Distance for motion)** computes the distance between the feature distributions of generated and ground-truth motions, reflecting realism at the distribution level.

As shown in Tab. 7, we present the formulas for the metrics used in the main text, with the definitions of relevant parameters as follows.

**Notation.** We denote the ground-truth motion sequence by $\mathbf{p}^{gt} = \{\mathbf{p}_t^{gt}\}_{t=1}^T$, and the $k$-th generated motion sequence by $\mathbf{p}^{(k)} = \{\mathbf{p}_t^{(k)}\}_{t=1}^T$, where $T$ is the prediction horizon and $K$ is the number of generated samples. Here, $\mathbf{p}_t \in \mathbb{R}^{J \times 3}$ represents the 3D skeleton at timestep $t$, with $J$ denoting the number of joints. For CMD, we compute the average displacement of all joints at frame $t$ as $M_t$, and the average displacement over the whole test set as $\overline{M}$. $(\mu, \Sigma)$ are the mean and covariance of extracted motion features used for FID calculation.

## A.7    COMPARISON WITH ANISOTROPIC NOISE.

Although our J-Noise shares similarities with SkeletonDiffusion's (Curreli et al., 2025) anisotropic noise, they differ fundamentally in design. SkeletonDiffusion treats the human skeleton as a pre-defined graph with fixed, manually specified joint-wise covariance. In contrast, J-Noise learns a dynamic scale $s_j = f_\theta(j, x_j^{1:H})$ directly from data, making the injected noise both joint-dependent and motion-dependent.

To evaluate the contribution of this proposed module, we conduct a detailed ablation study comparing the full model with two reduced variants: (1) **No J-Noise**, which adopts a uniform scalar noise schedule shared across all joints, and (2) **No Temporal**, which learns static per-joint noise scales but omits temporal motion cues.

Table 7: Evaluation metrics used for motion prediction.

| Metric | Formula |
|--------|---------|
| APD | $\dfrac{1}{K(K-1)}\sum_{i<j}\dfrac{1}{T}\sum_{t=1}^{T}\|\mathbf{p}_t^{(i)}-\mathbf{p}_t^{(j)}\|_2$ |
| ADE | $\dfrac{1}{T}\sum_{t=1}^{T}\|\hat{\mathbf{p}}_t-\mathbf{p}_t^{gt}\|_2$ |
| FDE | $\|\hat{\mathbf{p}}_T-\mathbf{p}_T^{gt}\|_2$ |
| CMD | $\sum_{t=1}^{T-1}(T-t)\,\|M_t-\bar{M}\|_1$ |
| FID* | $\|\mu_g-\mu_r\|_2^2+\mathrm{Tr}\Big(\Sigma_g+\Sigma_r-2(\Sigma_g\Sigma_r)^{1/2}\Big)$ |

The results in Tab. 8 highlight the importance of both joint-aware and temporally adaptive noise modeling. The No J-Noise variant exhibits the highest diversity (APD) but suffers from significantly degraded accuracy (ADE/FDE) and realism (FID), as the identical noise assignment fails to distinguish between highly dynamic limb joints and more stable trunk joints. Introducing joint-wise noise scaling (No Temporal) alleviates this issue and leads to measurable improvements, yet the model still underperforms compared to the full version.

In contrast, the Full J-Noise module—by adapting noise magnitude based on both joint identity and motion history, achieves the best performance across accuracy and physical realism metrics, while maintaining competitive diversity. These findings confirm that adaptive, temporally modulated joint-wise noise regulation is essential for generating accurate, coherent, and physically plausible motion.

Table 8: Detailed ablation of the Joint-Adaptive Noise Generator. The full model achieves the best accuracy and realism while maintaining competitive diversity.

| Method | ADE ↓ | FDE ↓ | FID ↓ | APD ↑ |
|--------|-------|-------|-------|-------|
| No J-noise | 0.339 | 0.454 | 0.088 | **7.243** |
| No temporal | 0.336 | 0.452 | 0.086 | 7.041 |
| Full J-Noise (ours) | **0.331** | **0.449** | **0.083** | 6.912 |

A.8 COMPARISON WITH STANDARD BONE-LENGTH CONSTRAINTS.

Although bone-length consistency is a commonly used constraint in human motion generation (Liang et al., 2024), our approach differs in how it is integrated into the diffusion model. Instead of predicting noise, the denoiser is designed to output the denoised human motion $\hat{y}_0$ at every diffusion timestep. After the initial encoding stage, all subsequent operations occur directly in the space of a noisy but valid 3D skeleton, which enables anatomical constraints to be applied at every stage of the denoising process.

Prior methods such as InterGen (Liang et al., 2024) enforce bone-length consistency only on the final reconstructed motion sequence, meaning the constraint influences the model once at the end of generation. In contrast, our step-wise formulation exposes the model to structural information during the entire denoising trajectory, encouraging globally consistent pose generation.

To assess the impact of this design, we replace our step-wise constraint with the traditional final-step constraint. As shown in Tab. 9, enforcing structure at every timestep yields consistently better accuracy and realism, demonstrating the clear benefit of integrating anatomical consistency throughout diffusion.

Table 9: Comparison with Standard Bone-Length Constraints.

| Constraint Timing | ADE ↓ | FDE ↓ | FID ↓ |
|---|---|---|---|
| Final step | 0.336 | 0.455 | 0.089 |
| Each step (Ours) | **0.331** | **0.449** | **0.083** |

## A.9 Disscussion of Learned Noise Scales

To investigate what is learned by the Joint-Adaptive Noise Generator, we visualize the learned joint-wise noise scales and find that they correlate with each joint's motion characteristics.

We categorized joints into three groups: Trunk (indices 0, 3, 6, 7, 8, 9, 10, 13), Knees & Elbows (1, 4, 11, 14), and Wrists & Feet (2, 5, 12, 15). We compared three settings:

- Ours: Initialized with increasing scales (1.0, 1.2, 1.4) and a learnable offset (clamped to $[-0.3, 0.3]$), encouraging higher variance for dynamic extremities.
- Uniform: Fixed scale of 1.0 for all joints.
- Reverse-scale: Initialized with decreasing scales (1.0, 0.8, 0.6) and a restricted learnable offset (clamped to $[-0.2, 0.2]$), forcing lower variance for extremities.

Analysis of Learned Scales: As shown in Tab. 10 and Fig. 8, our model learns to assign significantly higher noise scales to dynamic joints (e.g., Joint 15/R-Wrist: 1.41, Joint 5/L-Foot: 1.37) compared to stable trunk joints (e.g., Joint 0/Hip: 1.02). This aligns with the intuition that extremities have higher degrees of freedom and require more stochasticity. In contrast, the "Reverse-scale" setting forces the opposite pattern, suppressing noise in the limbs.

Performance Analysis: The impact of these noise distributions is clearly reflected in the quantitative results in Tab. 11. Ours achieves the best performance (ADE 0.331, FID 0.083), confirming that the learned kinematic-aware noise schedule effectively models human motion dynamics. Notably, reverse-scale performs significantly worse (ADE 0.349, FID 0.115) than even the Uniform baseline. This degradation indicates that incorrectly assigning low variance to dynamic joints (and high variance to stable ones) actively harms the generation process, further validating the necessity of our joint-adaptive design.

Table 10: The Learned Noise Scale per Joint in different noise settings on Human3.6M.

| Joint Index | 0 | 1 | 2 | 3 | 4 | 5 | 6 | 7 | 8 | 9 | 10 | 11 | 12 | 13 | 14 | 15 |
|---|---|---|---|---|---|---|---|---|---|---|---|---|---|---|---|---|
| Ours | 1.02 | 1.15 | 1.28 | 1.04 | 1.13 | 1.37 | 1.01 | 1.11 | 1.03 | 1.04 | 1.06 | 1.28 | 1.36 | 1.11 | 1.21 | 1.41 |
| Reverse | 0.98 | 0.84 | 0.73 | 1.00 | 0.87 | 0.75 | 1.05 | 1.09 | 1.07 | 1.11 | 1.07 | 0.83 | 0.71 | 1.06 | 0.85 | 0.73 |
| Uniform | 1.00 | 1.00 | 1.00 | 1.00 | 1.00 | 1.00 | 1.00 | 1.00 | 1.00 | 1.00 | 1.00 | 1.00 | 1.00 | 1.00 | 1.00 | 1.00 |

Table 11: Effect of different joint noise settings.

| Method | ADE ↓ | FDE ↓ | FID ↓ |
|---|---|---|---|
| Uniform | 0.339 | 0.454 | 0.088 |
| Reverse scale | 0.349 | 0.462 | 0.115 |
| Ours | **0.331** | **0.449** | **0.083** |

## A.10 Discussion on realism/accuracy and diversity of our model.

Tab. 3 shows that KinemaDiff achieves state-of-the-art accuracy and realism, while its APD score is slightly lower than several baselines. This phenomenon does not indicate a limitation of the noise generation module, but rather reflects a fundamental characteristic of enforcing anatomical consistency. Methods that report higher APD values often achieve diversity by producing motion samples that violate physical constraints, such as unrealistic bone stretching or implausible joint trajectories. In contrast, the incorporation of both Structure-Aligned Regularizer and the Joint-Adaptive Noise

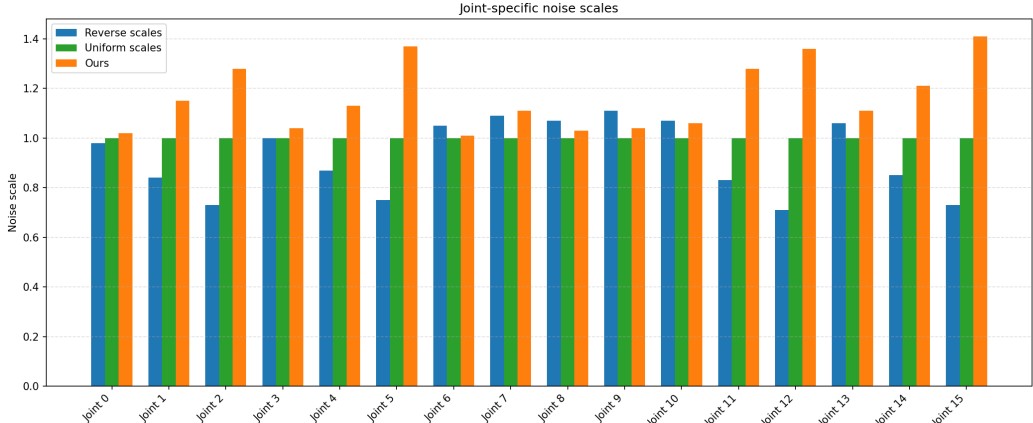

Figure 8: Visualization of the learned noise scales compared to Uniform and Reverse-scale baselines on Human3.6M. Our method automatically learns to assign larger noise scales to dynamic limb joints (indices 1, 2, 4, 5, 11, 12, 14, 15) to capture their higher degrees of freedom, while maintaining lower variance for the stable trunk. In contrast, the Reverse-scale setting enforces an unnatural distribution by suppressing noise in the extremities.

Generator restricts the generative distribution to anatomically feasible regions of the motion manifold. This "pruning" effect naturally leads to a more compact—yet physically valid—set of possible futures. As a result, KinemaDiff exhibits slightly reduced numerical diversity while achieving substantially stronger motion realism, as evidenced by its state-of-the-art FID performance. These findings highlight that our objective is not unrestricted diversity, but high-quality and physically plausible diversity, which more faithfully reflects the true distribution of human motion.

### A.11 EFFECTIVENESS OF EARLY-TIMESTEP STRUCTURAL CONSTRAINT.

We analyse the feasibility and benefits of applying structural constraints at early diffusion timesteps ($t \to 1$). While the predicted $\hat{x}_0$ at early stages (e.g., $t = 0.95$) exhibits high uncertainty and lacks high-frequency details compared to later stages ($t < 0.3$), applying structural constraints throughout the process is supported by the following rationale:

**Feasibility of Direct $x_0$ Prediction.** Unlike models that predict noise $\epsilon$, ours directly predicts the clean motion $x_0$ at every step. This means the network explicitly estimates a full skeleton even at high noise levels. While the *motion trajectory* (pose) might be uncertain or coarse at $t = 0.95$, the *skeletal structure* (bone lengths) is a deterministic property that should remain invariant.

*Visualization Analysis.* We added the output results for timesteps 1, 3, 6, and 8 (Fig. 9). Our visualizations show that even at early timesteps like 1 or 3, our model can produce a rough prediction. Specifically, for short-term predictions, the outputs at timesteps 1 or 3 are already close to the ground truth. Moreover, at these early timesteps, the top-performing hypotheses are fairly close to the ground truth, even for long sequence predictions. Therefore, within our direct prediction framework for $x_0$, imposing constraints at the early denoising stages is meaningful.

*Quantitative Analysis.* As shown in Tab. 12, our model produces reasonably structured outputs rather than random noise even at early steps. This further confirms that applying structural constraints is valid and feasible throughout the process.

**Reducing Search Space via Structural Constraints.** Applying constraints early acts as a strong geometric regularizer. By enforcing bone-length consistency, we decouple *structure* from *dynamics*. We essentially tell the model: "Even if you are unsure about the exact future pose, the output must be a valid human skeleton." This effectively prunes the search space, forcing the denoising trajectory to evolve strictly within the space of anatomically valid poses and preventing the model from wasting capacity on physically impossible distortions (e.g., stretched limbs).

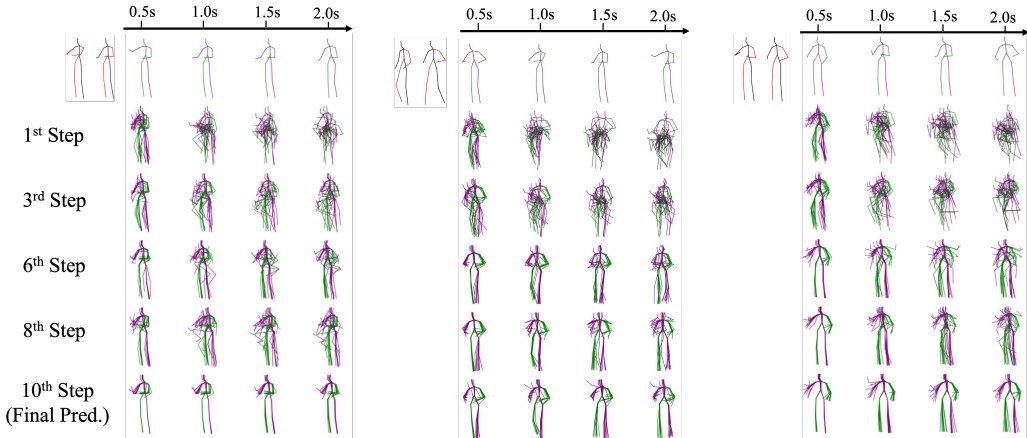

Figure 9: Performance across diffusion timesteps. The model achieves reasonable performance even at early denoising stages, with progressive improvement as the process proceeds to Step 10.

Table 12: Performance metrics across diffusion timesteps. The model achieves reasonable performance even at early denoising stages (Steps 1-3), with progressive improvement as the process proceeds to Step 10.

| Step | ADE ↓ | FDE ↓ | FID ↓ |
|---|---|---|---|
| 1st Step | 0.556 | 0.789 | 5.977 |
| 3rd Step | 0.443 | 0.632 | 3.728 |
| 6th Step | 0.358 | 0.490 | 0.763 |
| 8th Step | 0.336 | 0.455 | 0.139 |
| 10th Step (Ours) | 0.331 | 0.449 | 0.083 |

**Specialized Variance Schedule.** Enforcing constraints at very early timesteps could interfere with learning when the signal-to-noise ratio is extremely low. To address this, we intentionally adopt a variance scheduler that suppresses noise magnitude when $t$ approaches the highest-noise region. Specifically, as shown in Tab. 13, our noise variance ($\sqrt{1 - \bar{\alpha}_t}$) is significantly smaller than that of the Cosine and Sqrt schedules in early timesteps (e.g., 0.707 vs. 0.986 at timestep 1). This design is crucial: by reducing the noise level at early stages, the model can more easily recover meaningful skeletal structure from the noisy input, making it feasible to apply structural constraints effectively. In contrast, attempting to enforce constraints when predicting from near-complete noise (as in Cosine/Sqrt schedules) would be significantly more challenging and could hinder learning. Our approach ensures stable convergence and effective learning throughout the denoising process, as demonstrated in Tab. 4.

**Comparison with Late-Stage Constraint.** We conducted experiments applying the structural constraint only during the late denoising stages ($t < 0.2$ and $t < 0.3$) in Tab. 14, as is commonly done in other frameworks. In our direct prediction framework, imposing the constraint only on the last few timesteps still performs worse than applying it at every timestep. As shown in the table, restricting the constraint to late timesteps results in higher bone stretching (3.7 and 3.4 vs. 2.4) and degraded performance across all metrics compared to the full-process approach. This indicates that, for the direct prediction framework, waiting until the end to enforce structure is suboptimal.

Table 13: Comparison of Noise Schedules. The values represent the noise standard deviation ($\sqrt{1 - \bar{\alpha}_t}$) at each diffusion timestep.

| TimeStep | 1 | 2 | 3 | 4 | 5 | 6 | 7 | 8 | 9 | 10 |
|---|---|---|---|---|---|---|---|---|---|---|
| Cosine | 0.986 | 0.948 | 0.887 | 0.805 | 0.703 | 0.584 | 0.451 | 0.307 | 0.155 | 0.005 |
| Sqrt | 0.831 | 0.747 | 0.676 | 0.609 | 0.544 | 0.477 | 0.406 | 0.327 | 0.228 | 0.007 |
| Ours (Variance) | 0.707 | 0.650 | 0.588 | 0.523 | 0.454 | 0.383 | 0.309 | 0.233 | 0.156 | 0.079 |

Table 14: Comparison with Late-Stage Constraints.

| Constraint Timing | ADE ↓ | FDE ↓ | FID ↓ | Stretch ↓ |
|---|---|---|---|---|
| $t < 0.2$ | 0.336 | 0.458 | 0.113 | 3.7 |
| $t < 0.3$ | 0.335 | 0.452 | 0.106 | 3.4 |
| Ours (All steps) | **0.331** | **0.449** | **0.083** | **2.4** |

