# OpenReview forum: "KinemaDiff: Towards Diffusion for Coherent and Physically Plausible Human Motion Prediction"
_ICLR.cc/2026/Conference — ICLR 2026 Poster_

### Official Review · Reviewer_vAUr · 2025-10-23

**Soundness:** 3
**Presentation:** 3
**Contribution:** 2
**Rating:** 4
**Confidence:** 4

**Summary:**

The paper proposes a method to address stochastic human motion prediction, particularly by integrating anatomical consistency into the generative diffusion process. The paper proposes: (a) a Joint-Adaptive Noise Generator, which learns noise scheduler specifically tailored for different join, to take into account its characteristics in the human dynamic, (b) a Structure-Aligned Constraint Enforcer, which integrates the bone length information into the diffusion process by a training loss, enforcing that the avarage length of the bones in the prediction aligns with the prediction in the observed past. The paper compares with relevant state-of-the-art methods, showing promising results on AMASS and H3.6M.

**Strengths:**

- The idea of learning noise schedulers for different individual joints is interesting, and as far as I know, not yet explored
- The presentation is clear, and the paper is easy to follow.

**Weaknesses:**

- The "Structure-Aligned Constraint Enforcer" seems to be a simple loss that penalizes the error in the bones' length between the observed past and the prediction. Hence, its contribution seems a bit overstated in the paper, as well as the name is a bit deceptive. Also, in the ablation, it is not reported the case where Encoder and J-Noise are activated, but the Enforcer is not. This would be important to actually assess the contribution of this component.

- Although the paper introduces a loss specifically to take into account bone length, there are no metrics to measure this aspect.  This is surprising, as Skeleton Diffusion actually proposed to measure limb stretch and jittering across the prediction, since it has been observed that more diverse results correlate with unrealistic deformation of the skeleton.

- The paper does not include an analysis of what is learned by the Joint-Adaptive Noise Generator. I would find it interesting, as it can provide insights and interpretability in the proposed framework.

MINOR:
The work does not discuss/compare with [1,2]

[1]: Xu, S., Wang, Y. X., & Gui, L. Y. (2022, October). Diverse human motion prediction guided by multi-level spatial-temporal anchors. In European Conference on Computer Vision (pp. 251-269). Cham: Springer Nature Switzerland.

[2]: Xu, G., Tao, J., Li, W., & Duan, L. (2024, September). Learning semantic latent directions for accurate and controllable human motion prediction. In European Conference on Computer Vision (pp. 56-73). Cham: Springer Nature Switzerland.

**Questions:**

Currently, I slightly lean toward rejection: I find the proposed modification interesting, but it is also quite limited as a contribution, and some evidence is still missing to fully establish its validity. For the rebuttal, I'd like to see the following questions addressed:

1) Is it possible to clarify the "Structure-Aligned Constraint Enforcer" contribution, in light of the weaknesses above?
2) Would it be possible to see the metrics about limbs jittering and scratching for your comparison setting?
3) Would it be possible to include the ablation where only the Enforcer is removed?
4) A minor question lies in the nature of the choice of not relying on a latent diffusion, though the J-Noise module could also be applied there. Is there any reason or intuition behind this choice?

---

> ### Author Response · Authors · 2025-11-22
> **Response to Reviewer vAUr**
>
> We sincerely appreciate your detailed and insightful reviews. We have provided detailed responses to your comment and updated the relevant content in the revised manuscript, hoping our response can address your concerns.
>
> > **Q1: The "Structure-Aligned Constraint Enforcer" seems to be a simple loss that penalizes the error in the bones' length between the observed past and the prediction. Also, in the ablation, it is not reported the case where Encoder and J-Noise are activated, but the Enforcer is not.**
>
> Thank you for the comment. Our method is not merely a simple loss that penalizes bone-length errors. Following the suggestion raised by **Reviewer v3Dk Q2**, we implemented the conventional version—applying the bone-length loss only on the final predicted frame—and found that it brings less improvement on Human3.6M. This indicates that the key challenge is not the loss itself, but how it is integrated into the diffusion pipeline. Motivated by this, we designed a formulation that injects the constraint into each diffusion timestep, allowing the model to progressively regulate anatomical consistency throughout the denoising process. This design leads to clear performance gains, confirming that the contribution lies in how the constraint is incorporated rather than the loss form itself.
>
> Also, We have added the ablation where Encoder and J-Noise are enabled while the Enforcer is disabled on Human3.6M. The results further confirm the effectiveness of the Enforcer, showing that its contribution is not redundant with the other two components.
>
>
> We hope that this explanation and the provided ablation study satisfactorily address your concerns. Thanks for your valuable feedback! We have updated the manuscript to reflect these changes.
>
>
>
> | Constraint Timing | ADE ↓ | FDE ↓ | FID ↓ |
> |---------|-------|-------|-------|
> | Final step | 0.336 | 0.455 | 0.089 |
> | Each step  (Ours) | **0.331** | **0.449** | **0.083** |
>
>
> | Encoder | J-Noise | Align | APD ↑ | ADE ↓ | FDE ↓ | FID ↓ |
> | :---: | :---: | :---: | :---: | :---: | :---: | :---: |
> | - | - | - | **19.601** | 0.852 | 0.775 | 2.393 |
> | ✓ | - | - | 9.600 | 0.653 | 0.574 | 0.932 |
> | - | - | ✓ | 6.214 | 0.354 | 0.478 | 0.177 |
> | ✓ | - | ✓ | 7.243 | 0.339 | 0.454 | 0.088 |
> | ✓ | ✓ | - | 7.014 | 0.336 | 0.453 | 0.089 |
> | ✓ | ✓ | ✓ | 6.912 | **0.331** | **0.449** | **0.083** |
>
> > **Q2: Would it be possible to see the metrics about limbs jittering and scratching for your comparison setting?**
>
> Thank you for the suggestion. We computed mean **limb stretching** and **limb jitter** on Human3.6M.
>
> Our method achieves limb stretch of 2.42, significantly lower than SkeletonDiffusion's 3.90, mainly because our proposed Structure-Aligned Constraint Enforcer constrains the skeleton at each diffusion timestep, effectively reducing unrealistic deformation.
>
> For limb jittering, our model initially obtains 0.45. To further explore the flexibility of our Structure-Aligned Regularizer in enforcing temporal consistency, we adopted the same **jitter loss** as in SkeletonDiffusion as an additional constraint. Specifically, this loss quantifies the magnitude of bone length changes between adjacent frames. We calculate the bone length $L_t^j$ from 3D keypoints and define the jitter as the absolute difference $|L_{t+1}^j - L_t^j|$. The loss minimizes the squared difference between the predicted and ground-truth jitter values, constraining the generated bone dynamics to be temporally consistent with real motion.
>
> After adding this loss, the jitter metric
> significantly improves from 0.45 to 0.28. This demonstrates that our framework can readily incorporate such constraints to prevent excessive high-frequency shaking. Crucially, incorporating this loss does not compromise other metrics (ADE/FDE/FID remain stable), confirming that our framework can effectively balance structural validity and temporal smoothness.
>
> We hope that these additional metrics and the detailed comparison address your concerns regarding the physical realism of our method. Thanks for your valuable feedback! We have updated the manuscript to reflect these changes.
>
> | Method | Limb Stretch ↓ | Limb Jitter ↓ | ADE ↓ | FDE ↓ | FID ↓ |
> | :--- | :---: | :---: | :---: | :---: | :---: |
> | SkeletonDiffusion | 3.90 | 0.16 | 0.344 | 0.450 | 0.123 |
> | KinemaDiff (Ours) | **2.42** | 0.45 | **0.331** | 0.449 | **0.083** |
> | KinemaDiff + jitter loss | **2.42** | 0.28 | **0.331** | **0.447** | 0.084 |

---

> ### Author Response · Authors · 2025-11-22
> **Response to Reviewer vAUr**
>
> > **Q3: The paper does not include an analysis of what is learned by the Joint-Adaptive Noise Generator. I would find it interesting, as it can provide insights and interpretability in the proposed framework.**
>
> Thanks for the suggestion. We visualize the learned joint-wise noise scales in **Appendix Fig. 8** and find that they correlate with each joint's motion characteristics.
>
> To validate this, we categorized joints into three groups: **Trunk** (indices 0, 3, 6, 7, 8, 9, 10, 13), **Knees & Elbows** (1, 4, 11, 14), and **Wrists & Feet** (2, 5, 12, 15). We compared three settings:
>
> 1.  Ours: Initialized with increasing scales (1.0, 1.2, 1.4) and a learnable offset (clamped to $[-0.3, 0.3]$), encouraging higher variance for dynamic extremities.
> 2.  Uniform: Fixed scale of 1.0 for all joints.
> 3.  Reverse-scale: Initialized with decreasing scales (1.0, 0.8, 0.6) and a restricted learnable offset (clamped to $[-0.2, 0.2]$), forcing lower variance for extremities.
>
> Analysis of Learned Scales: As shown in **Table 1** below, our model learns to assign significantly higher noise scales to dynamic joints (e.g., **Joint 15/R-Wrist: 1.41**, **Joint 5/L-Foot: 1.37**) compared to stable trunk joints (e.g., **Joint 0/Hip: 1.02**). This aligns with the intuition that extremities have higher degrees of freedom and require more stochasticity. In contrast, the "Reverse-scale" setting forces the opposite pattern, suppressing noise in the limbs.
>
> Performance Analysis: The impact of these noise distributions is clearly reflected in the quantitative results in **Table 2** below. Ours achieves the best performance (ADE 0.331, FID 0.083), confirming that the learned kinematic-aware noise schedule effectively models human motion dynamics. Notably, Reverse-scale performs significantly worse (ADE 0.349, FID 0.115) than even the Uniform baseline. This degradation indicates that incorrectly assigning low variance to dynamic joints (and high variance to stable ones) actively harms the generation process, further validating the necessity of our joint-adaptive design.
>
> We hope that these visualizations and analyses provide the requested insights into the interpretability of our Joint-Adaptive Noise Generator. Thanks for your valuable feedback! We have updated the manuscript to reflect these changes.
>
> **Table 1: Average Noise Scale per Joint on Human3.6M**
> | Joint Index | 0    | 1    | 2    | 3    | 4    | 5    | 6    | 7    | 8    | 9    | 10   | 11   | 12   | 13   | 14   | 15   |
> |-------------|------|------|------|------|------|------|------|------|------|------|------|------|------|------|------|------|
> | Ours    | 1.02 | 1.15 | 1.28 | 1.04 | 1.13 | 1.37 | 1.01 | 1.11 | 1.03 | 1.04 | 1.06 | 1.28 | 1.36 | 1.11 | 1.21 | 1.41 |
> | Reverse | 0.98 | 0.84 | 0.73 | 1.00 | 0.87 | 0.75 | 1.05 | 1.09 | 1.07 | 1.11 | 1.07 | 0.83 | 0.71 | 1.06 | 0.85 | 0.73 |
> | Uniform | 1.00 | 1.00 | 1.00 | 1.00 | 1.00 | 1.00 | 1.00 | 1.00 | 1.00 | 1.00 | 1.00 | 1.00 | 1.00 | 1.00 | 1.00 | 1.00 |
>
> **Table 2: Performance Comparison on Human3.6M**
> | Method | ADE ↓ | FDE ↓ | FID ↓ |
> | :--- | :--- | :--- | :--- |
> | Uniform | 0.339 | 0.454 | 0.088 |
> | Reverse-scale | 0.349 | 0.462 | 0.115 |
> | Ours | **0.331** | **0.449** | **0.083** |
>
>
> > **Q4: A minor question lies in the nature of the choice of not relying on a latent diffusion, though the J-Noise module could also be applied there. Is there any reason or intuition behind this choice?**
>
> Thank you for the question. Our choice not to adopt latent diffusion stems from the design of our **direct diffusion** formulation, where each timestep explicitly predicts human motion. This allows us to directly apply geometric constraints (like bone lengths) in the data space at every step, which is harder to achieve in a compressed latent space. However, our J-Noise module is general and could potentially be adapted to latent diffusion models in future work. We have updated the manuscript to reflect these changes.
>
> > **Q5: Missing related work.**
>
> Thanks for pointing out the missing references. We have added citations and discussions for [1] and [2].
>
> 1. Xu et al. [1] propose an anchor-based generative framework that learns deterministic spatial-temporal anchors to capture multi-modal human motions, emphasizing diverse and controllable motion generation.
> 2. Xu et al. [2] introduce Semantic Latent Directions (SLD) to construct a constrained latent motion space, enabling more accurate and controllable predictions through lightweight semantic latent manipulations.
>
> We clarify that while they use anchor-based or latent direction methods, KinemaDiff uniquely integrates structural priors directly into the diffusion denoising process. We have updated the manuscript to reflect these changes.
>
>
>
> Reference:
>
> [1] Diverse human motion prediction guided by multi-level spatial-temporal anchors. ECCV 2022.
>
> [2] Learning semantic latent directions for accurate and controllable human motion prediction. ECCV 2024.

---

> > ### Comment · Reviewer_vAUr · 2025-11-24
> > **post rebuttal**
> >
> > I thank the authors for their replies to my concerns. I have carefully reviewed the other comments and the author's rebuttal, and at this time, I do not have any further questions. I look forward to discussing with other reviewers, after which I will enter my final assessment.

---

### Official Review · Reviewer_grBA · 2025-10-31

**Soundness:** 3
**Presentation:** 3
**Contribution:** 3
**Rating:** 6
**Confidence:** 3

**Summary:**

This paper introduces KinemaDiff, a novel diffusion-based framework for stochastic human motion prediction designed to improve physical plausibility and motion diversity. Conventional methods often suffer from anatomical inconsistencies and ignore heterogeneous joint dynamics. KinemaDiff addresses this by integrating two core modules directly into the diffusion process: (1) a Joint-Adaptive Noise Generator, which learns to inject instance-aware, heterogeneous noise for each joint based on its specific dynamics and motion history, and (2) a Structure-Aligned Constraint Enforcer, which preserves anatomical consistency by embedding skeletal topology (i.e., bone lengths) as an alignment loss during the denoising process. Experiments on the Human3.6M and AMASS datasets demonstrate that KinemaDiff achieves state-of-the-art performance, particularly in prediction accuracy (ADE/FDE) and realism (FID/CMD).

**Strengths:**

The paper's primary strength lies in its fundamental adaptation of the diffusion process itself, rather than merely modifying the denoiser's network architecture. It embeds kinematic and anatomical priors directly into the noising and denoising steps.

The Joint-Adaptive Noise Generator is a significant innovation. Moving beyond uniform or static anisotropic noise, this module learns a noise schedule conditioned on both the joint index and its specific motion history. This is more physically grounded, as different joints (e.g., wrist vs. hip) and different motions (e.g., walking vs. jumping) inherently have different degrees of freedom and stochasticity.

The Structure-Aligned Constraint Enforcer directly tackles a major failure mode of generative models, physical implausibility (e.g., stretching bones). By defining an explicit alignment loss based on bone lengths from observed motion, the model is strongly guided to produce anatomically consistent skeletons.

The method achieves state-of-the-art results on two standard benchmarks (Human3.6M and AMASS). The improvements in accuracy metrics (ADE/FDE) and, most notably, realism metrics (FID/CMD) are substantial, validating the effectiveness of the proposed components.

The ablation in Table 3 clearly demonstrates the contribution of each module. The Structure-Aligned module ("Align") provides the most significant boost in accuracy (ADE/FDE) and realism (FID), while the Joint-Adaptive Noise ("J-Noise") further improves these metrics, confirming their synergistic value.

**Weaknesses:**

The name "Structure-Aligned Constraint Enforcer" is a misnomer. The method described is a soft constraint or regularizer implemented via a loss function. It "encourages" anatomical consistency during training but does not "enforce" it, meaning it does not mathematically guarantee that the final output will be physically plausible.

The alignment loss is defined as the discrepancy between the average bone length over the observed history and the average bone length over the entire predicted future sequence. This is a very weak constraint. It allows for physically impossible motions (e.g., a bone shrinking in one frame and stretching in another) as long as the average length over the sequence is correct. A true physical constraint would require bone lengths to be (nearly) constant at every predicted frame.

Section 3.5 is confusing. It states, "at each timestep, after the initial encoder, we apply the same operation on $y_0$ to ensure that the human skeleton structure remains consistent". This implies the alignment loss is applied at every step $t$ of the denoising process. However, the overall loss function and the definition of the alignment loss based on the final prediction suggest it is only applied at the end of the training step. This ambiguity is critical for reproducibility.

The results (e.g., Table 1) show that while KinemaDiff excels in accuracy and realism, its diversity score (APD) is lower than several baselines. While this is a common trade-off, the paper could benefit from a deeper discussion on whether this is a fundamental consequence of enforcing stricter physical realism or a limitation of the current noise generation module.

**Questions:**

Regarding the "Constraint Enforcer": Did you experiment with a stronger, frame-wise alignment loss? For example, penalizing the L1 or L2 deviation from the observed bone length at each predicted future frame $f \in [1, F]$, rather than just penalizing the deviation of the sequence-level average? If so, how did this impact training stability and the final metrics?

Could you please clarify the exact mechanism described in Section 3.5? Is the alignment loss computed and backpropagated at every denoising timestep $t$ during training? Or is it only applied once to the final prediction $\hat{y}_0$, as suggested by Equation 12?

For the Joint-Adaptive Noise Generator, the function $f_{\theta}$ maps the joint index $j$ and its full history $x_j^{(1:H)}$ to a scaling factor $s_j$. Given that the paper describes this as "a few linear layers", how is the variable-length temporal history $x_j^{(1:H)}$ aggregated into a fixed-size vector to be processed by these layers?

The results show a trade-off between realism/accuracy and diversity (APD). In your view, is this lower diversity a necessary consequence of enforcing strict anatomical constraints (i.e., the model correctly "prunes" unrealistic but diverse motions), or do you see potential in the Joint-Adaptive Noise module to further enhance diversity while maintaining the high level of realism?

---

> ### Author Response · Authors · 2025-11-22
> **Response to Reviewer grBA**
>
> We sincerely appreciate your detailed and insightful reviews. We have provided detailed responses to your comment and updated the relevant content in the revised manuscript, hoping our response can address your concerns.
>
>
> > **Q1: The name "Structure-Aligned Constraint Enforcer" is a misnomer. The method described is a soft constraint or regularizer implemented via a loss function. It "encourages" anatomical consistency during training but does not "enforce" it, meaning it does not mathematically guarantee that the final output will be physically plausible.**
>
> Sorry for the imprecise naming. We agree that "Enforcer" implies a hard constraint, whereas our method functions as a soft regularizer during training. We have renamed the module to 'Structure-Aligned Regularizer' throughout the manuscript to accurately reflect its role in encouraging, rather than strictly enforcing, anatomical consistency. Thanks for your valuable feedback! We have updated the manuscript to reflect these changes.
>
> > **Q2: The average bone length is a weak constraint compared to frame-wise enforcement. Did you experiment with stronger frame-wise alignment losses (e.g., L1 or L2) to ensure constant bone lengths?**
>
> Thanks for the insightful suggestion. We agree that frame-wise enforcement is a stricter physical constraint. To investigate this, we implemented both L1 and L2 frame-wise bone-length penalties on Human3.6M.
>
> We found that the L1-based formulation was not suitable for this task, leading to significantly degraded performance (ADE 0.457). This instability likely arises because the L1 gradient is constant, which can be too aggressive for the delicate denoising process. The L2-based constraint effectively improves physical realism (lowest FID) by strictly enforcing geometry. However, this comes at the cost of slightly reduced prediction accuracy (ADE) compared to our average constraint, as the strict frame-wise penalty may over-constrain the motion manifold.
>
> We hope that this explanation and the provided ablation study address your concerns. Thanks for your valuable feedback! We have updated the manuscript to reflect these changes.
>
> | Loss variant | ADE ↓ | FDE ↓ | FID ↓ |
> |--------------|-------|-------|-------|
> | Frame-wise $L_1$ | 0.457 | 0.574 | 0.512 |
> | Frame-wise $L_2$ | 0.333 | 0.452 | **0.075** |
> | Average (Ours) | **0.331** | **0.449** | 0.083 |
>
>
> > **Q3: Section 3.5 is ambiguous: is the alignment loss applied at every timestep?**
>
> Thank you for pointing out this ambiguity. To clarify, the alignment loss is indeed computed and backpropagated at every diffusion timestep, not only at the end. Specifically, it is used in two places at each step:
>
> 1. Immediately after the initial encoder output.
> 2. On the predicted motion at every denoising timestep.
>
> This is made possible because our framework performs direct future-motion prediction at every timestep, allowing the alignment loss to be incorporated throughout the entire diffusion trajectory rather than only on the final output. Thanks for your valuable feedback! We have updated the manuscript to reflect these changes.
>
> > **Q4: The results (e.g., Table 1) show that while KinemaDiff excels in accuracy and realism, its diversity score (APD) is lower than several baselines. In your view, is this lower diversity a fundamental consequence of enforcing strict anatomical constraints (i.e., pruning unrealistic motions), or a limitation of the current noise generation module?**
>
> Thanks for raising this important point regarding the trade-off between accuracy/realism and diversity. We respectfully submit that in our context, the observed APD reflects a more **physically plausible** generative space rather than a limitation.
>
> Many methods (including our ablated baselines) can achieve high APD scores by generating diverse but physically unrealistic motions (e.g., bone stretching). By incorporating **kinematic-aware structural priors**, our model effectively "prunes" these anatomically impossible futures. While this naturally tightens the prediction distribution (resulting in a slightly lower APD), it ensures that the generated motions are much closer to the true distribution of human movements, as evidenced by our advanced ADE scores.
>
> Our goal is high-quality, physically plausible diversity rather than simple numerical diversity. We hope that this explanation addresses your concerns. Thanks for your valuable feedback! We have updated the manuscript to reflect these changes.

---

> > ### Author Response · Authors · 2025-11-22
> > **Response to Reviewer grBA**
> >
> > > **Q5: For the Joint-Adaptive Noise Generator, the function maps the joint index and its full history to a scaling factor. Given that the paper describes this as "a few linear layers", how is the variable-length temporal history aggregated into a fixed-size vector to be processed by these layers?**
> >
> > Thank you for the question. We apologize for the lack of clarity regarding this architectural detail.
> >
> > In our current experiments, we focus on tasks where the observation window is **fixed**, so the history is directly used. If handling **variable-length** histories is required, we can simply employ standard temporal modules (e.g., Transformer) to pool the history into a fixed-size feature vector before feeding it into the network.
> >
> > We hope that this explanation addresses your concerns. Thanks for your valuable feedback! We have updated the manuscript to reflect these changes.

---

### Official Review · Reviewer_v3Dk · 2025-11-01

**Soundness:** 2
**Presentation:** 3
**Contribution:** 2
**Rating:** 4
**Confidence:** 4

**Summary:**

This paper presents KinemaDiff, a diffusion-based framework for stochastic human motion prediction. The primary idea is to embed kinematic heterogeneity and skeletal consistency directly within the denoising process. Two components are introduced:

**Joint-Adaptive Noise Generator** — assigns heterogeneous Gaussian noise scales to each joint and dynamically modulates them using temporal features derived from the observed motion history.

**Structure-Aligned Constraint Enforcer** — enforces bone-length invariance at each denoising step based on topology extracted from past motion sequences.

Experiments on Human3.6M and AMASS demonstrate superior performance over recent baselines in accuracy and realism, supported by ablation studies and qualitative visualizations.

**Strengths:**

Strong state-of-the-art results on Human3.6M and AMASS in terms of accuracy and realism.

Effective integration of heterogeneous noise and skeletal constraint within the generation process, improving physical plausibility.

Comprehensive experiments, including ablations, quantitative benchmarks, and qualitative visualization, with consistent performance gains.

Process-level enforcement of anatomical constraints mitigates unrealistic poses that prior methods often handle via post-processing.

**Weaknesses:**

1. The Joint-Adaptive Noise Generator is quite similar to SkeletonDiffusion[1]’s anisotropic noise, with the main addition being a temporal feature to adjust the scale per joint.

2. The Structure-Aligned Constraint Enforcer uses the common bone-length consistency idea seen in prior motion generation work (e.g. [2]), mainly changing it to be applied during each diffusion step.

3. The gains in quantitative metrics are modest, and the visual results are only compared with CoMusion, which limits the strength of the realism claim.

[1]Nonisotropic Gaussian Diffusion for Realistic 3D Human Motion Prediction, CVPR25
[2]InterGen: Diffusion-based Multi-human Motion Generation under Complex Interactions, IJCV2024

**Questions:**

1. I’d be curious to see a clearer ablation of the J‑Noise design — for instance, what’s the gain if you just use per‑joint independent noise instead of uniform noise, and then how much extra does adding the temporal information actually help?

2. Can you explain a bit more how your Structure‑Aligned Constraint Enforcer is different from the usual bone‑length consistency losses used in motion generation? It might also be worth running a simple ablation to show its specific impact.

3. Would it be possible to add visual comparisons with a couple more baselines, like SkeletonDiffusion and BeLFusion, so the realism claim is backed by broader evidence?

---

> ### Author Response · Authors · 2025-11-22
> **Response to Reviewer v3Dk**
>
> We sincerely appreciate your detailed and insightful reviews. We have provided detailed responses to your comment and updated the relevant content in the revised manuscript, hoping our response can address your concerns.
>
> > **Q1: Similarity to SkeletonDiffusion[1]’s anisotropic noise, with the main addition being a temporal feature to adjust the scale per joint. The gain of just using per joint independent noise instead of uniform noise, and then how much extra does adding the temporal information actually help.**
> We apologize for the confusion. Although our J-Noise shares some similarities with SkeletonDiffusion's anisotropic noise, they are fundamentally different in terms of design and mechanism.
>
> **Static vs. Learnable Structures:** SkeletonDiffusion treats the human skeleton as a predefined graph whose joint-wise covariance is manually specified and fixed. J-Noise instead learns a scale $s_j=f_\theta(j, x_j^{1:H})$ from data, making the injected noise both **joint-dependent** and **motion-dependent**. In practice, dynamic joints such as hands/feet automatically obtain larger noise scales, while the pelvis/torso remains small. As detailed in our response to **Reviewer (vAUr) Q3** regarding *the interpretability of the Joint-Adaptive Noise Generator*, intentionally reversing this assignment noticeably degrades ADE/FID, confirming that the learned schedule captures motion semantics.
>
> We also provide the requested ablation on Human3.6M below. We compare our full model against two variants: (1) **No J-noise**, which uses a standard uniform scalar noise schedule across all joints, and (2) **No temporal**, which learns static per-joint noise scales but ignores temporal motion history.
>
> The results demonstrate the importance of each component. No J-noise yields the highest diversity (APD) but suffers from poor accuracy (ADE/FDE) and realism (FID), as it treats stable trunks and dynamic limbs identically. No temporal improves upon this by distinguishing between joints, yet still lags behind our full method. Our **Full J-Noise** generator, by dynamically adapting noise based on both joint type and real-time motion history, achieves the best performance in terms of accuracy and physical realism (lowest FID/ADE/FDE), proving that adaptive noise regulation is crucial for high-quality motion generation.
>
>
> We hope these analyses clarify the distinction from SkeletonDiffusion. Thanks for your valuable feedback! We have updated the manuscript to reflect these changes.
>
> | Variant | ADE ↓ | FDE ↓ | FID ↓ | APD ↑ |
> |---------|-------|-------|-------|-------|
> | No J-noise | 0.339 | 0.454 | 0.088 | **7.243** |
> | No temporal | 0.336 | 0.452 | 0.086 | 7.041 |
> | Full J-Noise (Ours) | **0.331** | **0.449** | **0.083** | 6.912 |
>
>
> > **Q2: How is Structure Aligned Constraint Enforcer different from the usual bone length consistency losses used in motion generation （InterGen[2]）? It might also be worth running a simple ablation to show its specific impact.**
>
> We apologize for the confusion. While we acknowledge that bone-length consistency is a well-established concept, our method fundamentally differs in how it is integrated into the diffusion process:
>
> 1. **Special Design for every step constraint:** To enable applying constraints at every timestep, we design the model to **directly output the predicted human motion at each step rather than predicting noise**. This means that after the initial encoder, the model operates on a noisy but valid 3D human skeleton. Working directly in this 3D space allows us to impose the desired physical constraints at every stage of the denoising process.
>
> 2. **Difference from InterGen:** Prior works like InterGen apply bone-length consistency only to the final denoised sequence. More specifically, they predict noise and impose the constraint on the motion sequence generated at the last diffusion timestep, so the constraint touches the model once per trajectory. In contrast, our denoiser is trained to predict a structurally consistent at all timesteps, making the model aware of anatomical constraints throughout the entire denoising process.
>
> 3. **Impact:** To quantify the difference, we conducted an experiment replacing the our Structure Aligned Constraint Enforcer with the traditional constraint (applied only to the final output) on Human3.6M. As shown in the table below, our full step-wise method achieves significantly better performance. These results demonstrate that continuously enforcing structure during diffusion is substantially more effective than the final-step constraint approach.
>
> We hope that this explanation and the provided ablation study address your concerns. Thanks for your valuable feedback! We have updated the manuscript to reflect these changes.
>
>
> | Constraint Timing | ADE ↓ | FDE ↓ | FID ↓ |
> |---------|-------|-------|-------|
> | Final step | 0.336 | 0.455 | 0.089 |
> | Each step  (Ours) | **0.331** | **0.449** | **0.083** |

---

> > ### Author Response · Authors · 2025-11-22
> > **Response to Reviewer v3Dk**
> >
> > > **Q3: Would it be possible to add visual comparisons with a couple more baselines, like SkeletonDiffusion and BeLFusion, so the realism claim is backed by broader evidence.**
> >
> > Thanks for the suggestion. We have added relevant qualitative comparisons in the Appendix. These visualizations demonstrate that KinemaDiff significantly reduces bone stretching (highlighted by shorter limb error bars) observed in baselines. We have updated the manuscript to reflect these changes.
> >
> >
> >
> >
> >
> > Reference:
> >
> > [1] Nonisotropic Gaussian Diffusion for Realistic 3D Human Motion Prediction, CVPR25.
> >
> > [2] InterGen: Diffusion-based Multi-human Motion Generation under Complex Interactions, IJCV2024

---

> > > ### Comment · Reviewer_v3Dk · 2025-11-25
> > > **For Q2.1**
> > >
> > > Thank you for the authors' response. The additional experiments help to reduce my concerns. Regarding the "predicted human motion at each step", I wonder whether reasonable motion can be predicted in the very early steps. In my understanding, from a stage of almost pure noise $x_t$ (e.g., assuming $t=1 \rightarrow 0$; at $t=0.95$ or $t=0.9$), directly predicting $x_0$ may not be very accurate. Based on experience, we usually apply constraints to $x_0$ only when $t < 0.3 \ or \  0.2$. I am curious about how the authors view this issue.

---

> > > > ### Author Response · Authors · 2025-11-26
> > > > **Response to Reviewer v3Dk**
> > > >
> > > > > **Additional Question: Regarding the "predicted human motion at each step", I wonder whether reasonable motion can be predicted in the very early steps. In my understanding, from a stage of almost pure noise $x_t$ (e.g., assuming $t=1 \to 0$; at $t=0.95$ or $t=0.9)$, directly predicting $x_0$ may not be very accurate. Based on experience, we usually apply constraints to $x_0$ only when $t < 0.3$ or $0.2$. I am curious about how the authors view this issue.**
> > > >
> > > > We thank the reviewer for this insightful question regarding the timing of constraints. You are right that at early stages (e.g., $t=0.95$), the predicted $\hat{x}_0$ has high uncertainty and lacks high-frequency details compared to later stages ($t < 0.3$). However, we choose to apply structural constraints throughout the process based on the following logic:
> > > >
> > > > 1.  **Direct $x_0$ Prediction and Empirical Evidence.**
> > > >     Unlike models that predict noise $\epsilon$, KinemaDiff directly predicts the clean motion $x_0$ at every step. This means the network explicitly estimates a full skeleton even at high noise levels. While the *motion trajectory* (pose) might be uncertain or coarse at $t=0.95$, the *skeletal structure* (bone lengths) is a deterministic property that should remain invariant.
> > > >
> > > >     **Visualization Analysis:** We added the output results for timesteps 1, 3, 6, and 8  (**Fig. 9 in the appendix**). Our visualizations show that even at early timesteps like 1 or 3, our model can produce a rough prediction. Specifically, for short-term predictions, the outputs at timesteps 1 or 3 are already close to the ground truth. Moreover, at these early timesteps, the top-performing hypotheses are fairly close to the ground truth, even for long sequence predictions. Therefore, within our direct prediction framework for $x_0$, imposing constraints at the early denoising stages is meaningful.
> > > >
> > > >     **Quantitative Analysis:** As shown in the table below, our model produces reasonably structured outputs rather than random noise even at early steps. This further confirms that applying structural constraints is mathematically valid and feasible throughout the process.
> > > >
> > > >     | | ADE | FDE | FID |
> > > >     |---|---|---|---|
> > > >     | Step1 | 0.556 | 0.789 | 5.977 |
> > > >     | Step3 | 0.443 | 0.632 | 3.728 |
> > > >     | Step6 | 0.358 | 0.490 | 0.763 |
> > > >     | Step8 | 0.336 | 0.455 | 0.139 |
> > > >     | Step10 (Ours) | 0.331 | 0.449 | 0.083 |
> > > >
> > > > 2.  **Reducing Search Space via Structural Constraints.**
> > > >     Applying constraints early acts as a strong geometric regularizer. By enforcing bone-length consistency, we decouple *structure* from *dynamics*. We essentially tell the model: "Even if you are unsure about the exact future pose, the output must be a valid human skeleton." This effectively prunes the search space, forcing the denoising trajectory to evolve strictly within the space of anatomically valid poses and preventing the model from wasting capacity on physically impossible distortions (e.g., stretched limbs).
> > > >
> > > > 3.  **Specialized Variance Schedule:** We acknowledge the reviewer's concern that enforcing constraints at very early timesteps could interfere with learning when the signal-to-noise ratio is extremely low. To address this, we intentionally adopt a variance scheduler that **suppresses noise magnitude when $t$ approaches the highest-noise region**. Specifically, as shown in the table below, our noise variance ($\sqrt{1 - \bar{\alpha}_t}$) is significantly smaller than that of Cosine and Sqrt schedules in early timesteps (e.g., 0.707 vs. 0.986 at timestep 1). This design is crucial: by reducing the noise level at early stages, the model can more easily recover meaningful skeletal structure from the noisy input, making it feasible to apply structural constraints effectively. In contrast, attempting to enforce constraints when predicting from near-complete noise (as in Cosine/Sqrt schedules) would be significantly more challenging and could hinder learning. Our approach ensures stable convergence and effective learning throughout the denoising process.
> > > >
> > > >     **Comparison of Noise Variance Schedules:**
> > > >     | TimeStep | 1 | 2 | 3 | 4 | 5 | 6 | 7 | 8 | 9 | 10 |
> > > >     | :--- | :---: | :---: | :---: | :---: | :---: | :---: | :---: | :---: | :---: | :---: |
> > > >     | Cosine | 0.986 | 0.948 | 0.887 | 0.805 | 0.703 | 0.584 | 0.451 | 0.307 | 0.155 | 0.005 |
> > > >     | Sqrt | 0.831 | 0.747 | 0.676 | 0.609 | 0.544 | 0.477 | 0.406 | 0.327 | 0.228 | 0.007 |
> > > >     | Ours (Variance) | 0.707 | 0.650 | 0.588 | 0.523 | 0.454 | 0.383 | 0.309 | 0.233 | 0.156 | 0.079 |
> > > >
> > > >     **Performance Comparison:**
> > > >     | Scheduler | APD $\uparrow$ | ADE $\downarrow$ | FDE $\downarrow$ | FID $\downarrow$ |
> > > >     | :--- | :---: | :---: | :---: | :---: |
> > > >     | Sqrt | 6.837 | 0.342 | 0.457 | 0.108 |
> > > >     | Cosine | 7.213 | 0.365 | 0.478 | 0.178 |
> > > >     | Variance (Ours) | 6.912 | **0.331** | **0.449** | **0.083** |

---

> > > > > ### Comment · Reviewer_v3Dk · 2025-11-27
> > > > >
> > > > > Thank you for the author's further and detailed clarification. I will update my score.

---

> > > > ### Author Response · Authors · 2025-11-26
> > > > **Response to Reviewer v3Dk**
> > > >
> > > > 4.  **Comparison with Late-Stage Constraints:** To directly address your query, we conducted additional experiments where we applied the structural constraint only during the late denoising stages ($t < 0.2$ and $t < 0.3$), as is common in other frameworks.
> > > >     In our direct prediction framework, we found that imposing the constraint only on the last few timesteps still performs worse than applying it at every timestep. As shown in the table below, restricting the constraint to late timesteps results in higher bone stretching (3.7 and 3.4 vs. 2.4) and degraded performance across all metrics compared to our full-process approach. This confirms that for our direct prediction framework, waiting until the end to enforce structure is suboptimal.
> > > >
> > > >     | Constraint Timing | ADE $\downarrow$ | FDE $\downarrow$ | FID $\downarrow$ | Stretch $\downarrow$ |
> > > >     | :--- | :---: | :---: | :---: | :---: |
> > > >     | $t < 0.2$ | 0.336 | 0.458 | 0.113 | 3.7 |
> > > >     | $t < 0.3$ | 0.335 | 0.452 | 0.106 | 3.4 |
> > > >     | Ours (All steps) | **0.331** | **0.449** | **0.083** | **2.4** |
> > > >
> > > > We hope that this explanation, along with the provided analysis and additional experiments, addresses your concerns. Thanks for your valuable feedback! We have updated the manuscript to reflect these changes.

---

### Author Response · Authors · 2025-11-22
**General Response**

We sincerely appreciate all reviewers and community members for their efforts in evaluating the paper and writing suggestions that greatly help us improve the work！ Please find our responses to your individual questions below. We look forward to discussing any issues further should you have any follow-up concerns!

---

### Author Response · Authors · 2025-11-30
**Brief summarization of discussion process prior to the score reversion. (Part #2/2)**

> **Reviewer v3Dk**

This reviewer's concerns focused on three points: (1) questions about how J-Noise differs from SkeletonDiffusion and requesting step-by-step ablations to separate the contributions of joint-wise noise and temporal modulation; (2) our Structure-Aligned Constraint Enforcer appeared similar to the common bone-length consistency losses used in prior motion generation work, asking for clarification and a simple ablation; and (3) insufficient visual evidence without comparisons to SkeletonDiffusion and BeLFusion.

In our rebuttal, we clarified that J-Noise differs fundamentally by learning joint- and motion-dependent noise scales, dynamically adjusting based on real-time motion history rather than relying on a fixed covariance graph. Ablations confirm that removing J-Noise or ignoring temporal history degrades accuracy (ADE/FDE) and realism (FID), demonstrating the necessity of both components.
For the Structure-Aligned Constraint Enforcer, we explained that unlike prior works (e.g., InterGen) which apply bone-length constraints only on the final output, our method enforces anatomical consistency at each diffusion timestep. This continuous, step-wise integration yields significantly better performance, as verified by ablation comparing final-step versus per-step enforcement.
Overall, these analyses clarify both the novelty and impact of our components, supporting their effectiveness in producing accurate and physically plausible human motion.

After a follow-up question about the validity of early-step constraints, we further clarified this with visualizations, quantitative trends, and our noise scheduling design. ***The reviewer acknowledged these results resolved their initial doubts and raised their rating from "4: marginally below the acceptance threshold" to "6: marginally above the acceptance threshold".***

> **Reviewer grBA**

This reviewer highlighted several strengths: the method fundamentally adapts the diffusion process by embedding kinematic and anatomical priors; the Joint-Adaptive Noise Generator is a notable innovation that models joint- and motion-specific stochasticity; the Structure-Aligned Constraint Enforcer reduces physically implausible motions; and the approach achieves state-of-the-art accuracy and realism, with ablations confirming both modules’ contributions.
The main concerns focused on the naming and clarity of the structural module, the strength and formulation of the constraint, the application of the structural loss across diffusion timesteps, the handling of temporal history in the noise generator, and the lower APD compared to some baselines.

In our rebuttal, we addressed these concerns as follows: We renamed the "Structure-Aligned Constraint Enforcer" to "Structure-Aligned Regularizer" to accurately reflect its role as a soft anatomical prior. We conducted additional experiments comparing frame-wise L1/L2 penalties and found that L1 causes instability while L2 over-constrains motion, confirming that our average-length formulation achieves trade-off between realism and accuracy. Furthermore, we clarified that the structural regularizer is applied at every diffusion timestep, enabled by our direct prediction framework. Regarding the Joint-Adaptive Noise Generator, we explained that  current task uses a fixed-length observation window, and for variable-length histories, standard temporal pooling (e.g., Transformer) can be applied. Finally, we clarified that the slightly lower APD is expected, as our model suppresses anatomically implausible motions, ensuring physically meaningful diversity rather than artificially inflated scores.

> **Reviewer vAUr**

This reviewer found the idea of learning joint-wise noise scales interesting and novel within the context of motion diffusion, and appreciated the clarity of our presentation.
The concerns mainly centered on the simplicity of the proposed structural constraint, the lack of certain ablations and additional metrics (jitter and stretch), and the interpretability of what the Joint-Adaptive Noise Generator actually learns.

In our rebuttal, we clarified that the contribution of the Structure-Aligned Regularizer does not lie in introducing a new form of bone-length loss, but in the way it is incorporated into every denoising step of our direct diffusion formulation, which fundamentally changes how structural priors influence the generation trajectory. We also provided the missing analyses, including the limb stretch and joint jitter metrics, ablations on noise-scale initialization strategies, and visualizations showing how the learned joint-adaptive scales correlate with motion dynamics. These results directly address the reviewer’s questions and consistently support the effectiveness of our design.

***Reviewer vAUr is satisfied with the rebuttal: they thanked us for the clarifications, noted that they have no further questions. They plan to discuss with the other reviewers before finalizing their assessment.***

---

### Author Response · Authors · 2025-11-30
**Brief summarization of discussion process prior to the score reversion. (Part #1/2)**

Dear Area Chair, please allow us to briefly summarize our discussion process with the reviewers prior to the score reversion.

Initially, our submission received one "6: marginally above the acceptance threshold" (Reviewer **grBA**) and two "4: marginally below the acceptance threshold" scores (Reviewer **v3Dk** and Reviewer **vAUr**). During the discussion phase, the two reviewers who initially scored "4: marginally below the acceptance threshold" acknowledged our rebuttal and found that we satisfactorily addressed their concerns. Following further discussion, Reviewer **v3Dk** raised their score from 4 to 6, while Reviewer **vAUr** indicated that they would provide their final assessment after discussion with other reviewers.

Before the official score reversion, we had successfully addressed all reviewers' concerns. Reviewer **v3Dk** had already raised their score from 4 to 6, and Reviewer **vAUr** also stated they would provide their final assessment after discussion with other reviewers. At that point, our scores stood at 6, 6, and 4.

The following is a brief summary of our discussions with the reviewers:

---

### Meta-Review · Area_Chair_WPTK · 2026-01-06

**Summary:**

This paper addresses the task of Human Motion Prediction (HMP). Previous methods suffer from drawbacks including inconsistent skeletions and physical implausibility. In this paper, two modules are designed: one better aligns with human dynamics by separately treating different joints, and the other enhances physical feasibility by utilizing bone length information as constraints.

Three reviewers provided detailed comments and suggestions for the paper. They consider that its main strengths include the interesting and effective module that learns noise schedulers for individual joints, the clear presentation and logic, and comprehensive experiments with consistent performance gains. Their concerns mainly include questions regarding the contribution of the modules, insufficient ablation experiments, and limited innovation.

The authors provided a very detailed rebuttal to the reviewers' comments, including many additional supplementary quantitative and qualitative experimental results. Two of the reviewers responded again, acknowledging the authors' rebuttal.

Overall, the paper demonstrates coherent logic and solid experimentation, with an initial rating of 6, 4, 4 (the corresponding confidence being 3, 4, 4). The authors provided detailed rebuttal, which reflects their dedication to the work and their commitment to improving its quality. Their efforts are acknowledged by 2 reviewers who initially assigned a score of 4, so it is reasonable to infer that reviewers would have assigned a final rating of 6, 6, 6 if the discussion had been continued. Considering that Human Motion Prediction is a relatively specialized field, I am inclined to recommend accepting this paper.

**Reviewer Concerns:**

Regarding the specific questions raised by reviewer1 on presentation, I find the responses to be quite detailed and likely to effectively address the concerns. As for other content-related issues, in personal opinion, the authors' explanations are also reasonable.

Reviewer 2 have explicitly indicated that the score will be updated. Based on the discussion process, it is also evident that the authors' responses have been exceptionally detailed, specific, and thoughtful. Therefore, in personal opinion, the reviewer’s concerns have been addressed.

Reviewer 3 stated no further questions but will await discussions with other reviewers before finalizing the score. This suggests that the authors’ responses have addressed the majority of Reviewer 3’s concerns.

**Reviewer Scores:**

In personal opinion, Reviewer 1's final score will remain a 6, as the authors provided detailed explanations for each of the reviewer’s questions and generally addressed the concerns raised.

In personal opinion, Reviewer 2's final score will be raised to a 6. The reviewer indicated that this was the score prior to any revisions, and also stated in their response to the authors' rebuttal that they would raise the score.

In personal opinion, Reviewer 3's final score will likely be raised to a 6, or at least remain a 4. After reading the authors' rebuttal, the reviewer stated that they had no further questions and would decide on a final score after discussing with the other reviewers. This comment was made before Reviewer 2 indicated their intention to raise the score. Therefore, if Reviewer 3 sees that the other two reviewers are likely to assign scores of 6, they may also raise their score accordingly.

---

### Decision · Program_Chairs · 2026-01-26

Accept (Poster)